# Near-strain-free anode architecture enabled by interfacial diffusion creep for initial-anode-free quasi-solid-state batteries

Kwang Hee Kim[1,4], Myung-Jin Lee[2,4], Minje Ryu[1,4], Tae-Kyung Liu[1], Jung Hwan Lee[1], Changhoon Jung[3], Ju-Sik Kim [2] ✉ & Jong Hyeok Park [1] ✉

Anode-free (or lithium-metal-free) batteries with garnet-type solid-state electrolytes are considered a promising path in the development of safe and high-energy-density batteries. However, their practical implementation has been hindered by the internal strain that arises from the repeated plating and stripping of lithium metal at the interlayer between the solid electrolyte and negative electrode. Herein, we utilize the titanium nitrate nanotube architecture and a silver-carbon interlayer to mitigate the anisotropic stress caused by the recurring formation of lithium deposition layers during the cycling process. The mixed ionic-electronic conducting nature of the titanium nitrate nanotubes effectively accommodates the entry of reduced Li into its free volume space via interfacial diffusion creep, achieving near-strain-free operation with nearly tenfold volume suppressing capability compared to a conventional Cu anode counterpart during the lithiation process. Notably, the fabricated $Li_{6.4}La_3Zr_{1.7}Ta_{0.3}O_{12}$ (LLZTO)-based initial-anode-free quasi-solid-state battery full cell, coupled with an ionic liquid catholyte infused high voltage $LiNi_{0.33}Co_{0.33}Mn_{0.33}O_2$-based cathode with an areal capacity of $3.2\ mA\ cm^{-2}$, exhibits remarkable room temperature (25 °C) cyclability of over 600 cycles at $1\ mA\ cm^{-2}$ with an average coulombic efficiency of 99.8%.

The advancements in lithium-ion battery (LIB) technologies have brought groundbreaking changes to the automotive industry, spurring competition in the development of more sustainable and efficient electric vehicles (EVs)[1–3]. However, commercial LIBs still encounter many safety challenges associated with the use of flammable liquid electrolytes as well as technical difficulties caused by a constant volume change during charging/discharging[4,5]. In this regard, solid-state batteries (SSBs) have been touted as a promising solution owing to their enhanced safety and long-term cycle stability[6–9]. The use of nonflammable solid electrolytes (SEs) can effectively reduce the risk of fire hazards in the event of a short circuit or cell damage[10,11]. However, SSBs still face significant issues with cell volume change (equivalent to

cell thickness change), as several studies have indicated severe volume expansion/contraction during the charging/discharging process[12–14]. In particular, SSBs with anode-free configurations have been shown to be more susceptible to volumetric change due to repeated Li plating/stripping in the interlayer between the SE and current collector during the cycling process[15–17]. Recent advancements in the performance of garnet-type SE-based quasi-all-solid-state batteries (ASSBs), such as those seen in our own report of an Ag-coated Ag-C interlayer, have effectively mitigated dendrite penetration by inducing Li deposition at the interface between the interlayer and current collector[18]. Nonetheless, in our study, the cell experienced considerable volume change during the charging/

[1]Department of Chemical and Biomolecular Engineering, Yonsei University, 50 Yonsei-ro, Seodaemun-gu, Seoul 03722, Republic of Korea. [2]Battery Material TU, Samsung Advanced Institute of Technology, 130, Samsung-ro, Yeongtong-gu, Suwon-si, Gyeonggi-do 16678, Republic of Korea. [3]Analytical Engineering Group, Samsung Advanced Institute of Technology, 130, Samsung-ro, Yeongtong-gu, Suwon-si, Gyeonggi-do 16678, Republic of Korea. [4]These authors contributed equally: Kwang Hee Kim, Myung-Jin Lee, Minje Ryu. ✉e-mail: jusik.kim@samsung.com; lutts@yonsei.ac.kr

discharging cycle, which could result in significant internal stress if incorporated into a stacked cell configuration[18].

In the effort to develop strategies to solve this problem, the use of a mixed ionic and electronic conductor (MIEC)-based 3D tubular anode has proven effective in minimizing volume change during Li plating/stripping in polymer SE-based SSBs[19]. The MIEC-based 3D tubular anodes provide diffusion channels across their tubular network via the interfacial-diffusional Coble creep mechanism, which helps in the uniform deposition of Li inside the 3D architecture and minimizes the volume change during cycling. Additionally, the utilization of 3D porous ion-conductive hosts is reported to be effective in averting dendrite formation and volume change in garnet-type SE-based solid-state Li metal batteries[20,21]. These findings suggest that designing an interfacial structure that enables Li deposition in free volume spaces can help minimize volume changes during the cycling process, thereby reducing internal strain and preventing disintegration of the SE and negative electrode layer[22–24].

Herein, we propose a near-strain-free anode architecture for garnet-type $Li_{6.4}La_3Zr_{1.7}Ta_{0.3}O_{12}$ (LLZTO)-based anode-free (or Li-free) SSBs using vertically oriented titanium nitride nanotubes (TiN NTs). TiN NTs have high electrical conductivity and appropriate Li solubility at the TiN/$Li_{bcc}$ interface, allowing for stable movement of Li inside the nanotube structure[25,26]. Additionally, an Ag-C interlayer is embedded between the LLZTO and TiN NT architecture to facilitate the rapid transport of the reduced Li metal towards the TiN NT, thereby promoting uniform Li deposition within the TiN NT[27–29]. As depicted in Fig. 1a and b, the Ag-C-coated TiN NT anode effectively compensates for the volume change by

storing Li inside the voids of the nanotubes during the charging/discharging process. In contrast, in a conventional anode-free configuration, Li is deposited between the Ag-C and copper current collector interface, which can culminate in internal stress buildup. Furthermore, we explore the Li diffusion mechanism within the TiN NT array during cell operation to better understand Li transport at a macroscopic level. Finally, the near-strain-free TiN NT anode is applied to an anode-free quasi-solid-state battery (AFSSB) full cell comprising an ionic liquid (IL)-based electrolyte infused high-voltage cathode and pre-lithiated Ag-C-coated LLZTO. The quasi-solid-state battery with the IL additive enhanced the solid-solid interfacial contact and reduced interfacial impedance between the solid electrolyte and cathode[30,31]. Remarkably, compared to the copper current collector, the fabricated AFSSB full cell exhibits a tenfold suppression in volume expansion during the charging process and delivers excellent cycling stability for 600 cycles at a current density of 1.6 mA cm$^{-2}$ at room temperature (25 °C).

## Results

### Morphological characterization and lithiation behaviour analysis of the TiN NT architecture

The TiN NT architecture was fabricated via a two-step anodization process of metallic Ti foil, followed by annealing under NH$_3$ conditions[32,33]. As shown in Supplementary Figs. 1 and 2, the prepared TiN NTs had a pore size of approximately 100 nm and a thickness of 15 μm and showed a uniform morphology regardless of the initial thickness of the Ti foil. The successful conversion to the TiN phase was confirmed by the absence of a TiO$_2$ peak in the X-ray diffraction (XRD)

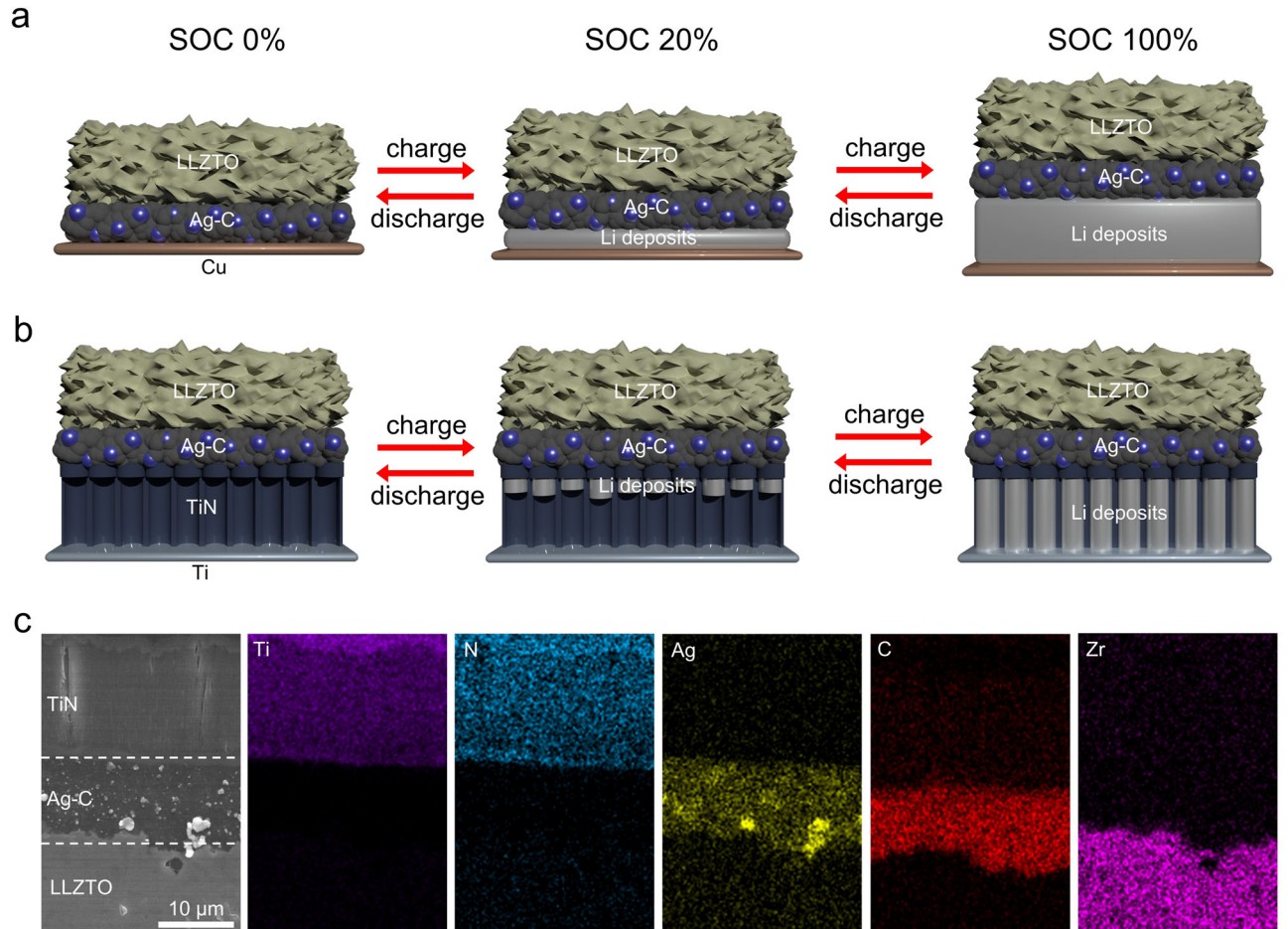

**Fig. 1 | Illustration of the near-strain-free operation of the TiN NT-incorporated garnet-type SE-based AFSSB. a** Schematic of the conventional AFSSB configuration (with Cu as an anode) during the charging/discharging process. **b** Schematic of the TiN NT-incorporated AFSSB during the charging/discharging process. **c** Cross-sectional SEM image and corresponding EDS images of the TiN NT-incorporated AFSSB.

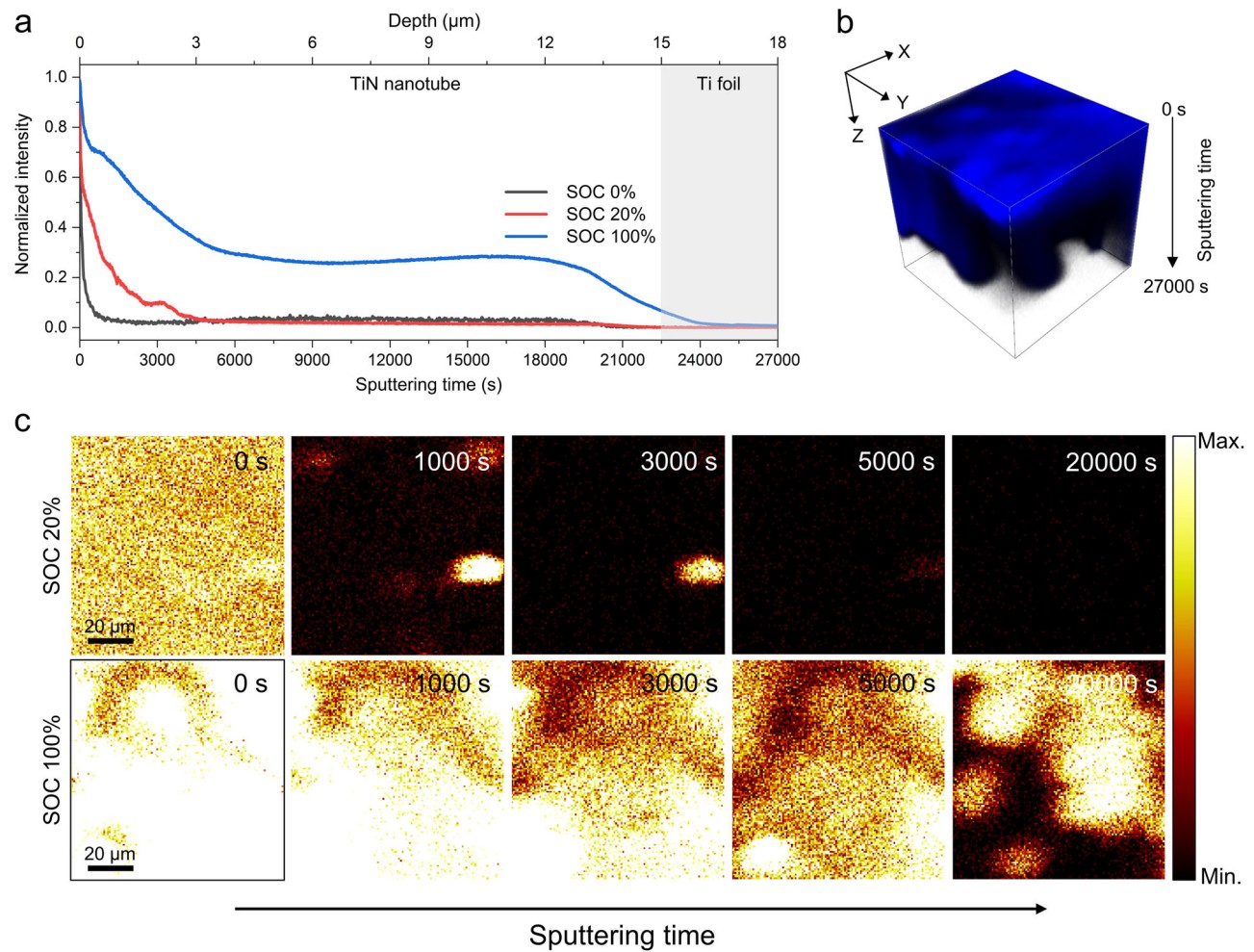

**Fig. 2 | Li deposition behaviour in the TiN NT at different SOCs. a** Normalized TOF-SIMS depth profile of Li⁺ secondary ions for TiN NTs at various SOCs. **b** A 3D rendering of the Li⁺ secondary ion depth profiles in (**a**) at 100% SOC. **c** TOF-SIMS 2D chemical mapping of Li⁺ secondary ion signals at various sputtering times.

pattern (Supplementary Fig. 3). Then the prepared TiN NT architecture was incorporated into the Ag-C-coated LLZTO pellet via cold isostatic pressing (CIP) under 250 MPa (Fig. 1c and Supplementary Fig. 4). The use of the Ag-C interlayer promoted the stable plating of Li metal within the TiN NT cavities and ensured continuous interfacial contact between the SE and TiN NT. To investigate the Li diffusion mechanism across the TiN NT architecture, a full cell was assembled using TiN NTs as the anode and an IL-based electrolyte-infused $LiNi_{0.33}Co_{0.33}Mn_{0.33}O_2$ (NCM333) as the cathode. The maximum areal capacity of the TiN NT anode based on the porosity of the TiN NT and the capacity of Li metal was calculated as 2.473 mAh/cm² (Supplementary Fig. 5). To prevent Li overflow and plating onto the TiN NT surface, we designed the capacity of the NCM333 cathode not to exceed the maximum capacity of the TiN NT anode. Using the fabricated full cell, we conducted a time-of-flight secondary ion mass spectrometry (TOF-SIMS) analysis at two different states of charge (SOCs), i.e., 20% SOC and 100% SOC, during the charging process (Fig. 2). The 20% SOC sample was prepared by charging the full cell to a capacity cut-off of 0.64 mAh cm⁻², calculated based on the theoretical areal capacity of the NCM333 (3.2 mAh cm⁻²), and the 100% SOC sample was prepared by completely lithiating the full cell to a cut-off voltage of 4.3 V (Supplementary Fig. 6). Figure 2a shows the normalized depth profile of the Li⁺ secondary ion signal, representing the relative intensity of the deposited Li metal along the depth of the TiN NTs for each charged sample. Beyond a sputtering time of approximately 22,500 seconds, the signals of $NH_3^+$ and Li⁺ ions disappeared,

leaving only the Ti⁺ secondary ion signal, indicating that the etched TiN NT reached the Ti foil (current collector) (Supplementary Fig. 7a, b). It is worth noting that as the sputtering time increased, the intensity of the Li⁺ secondary ion signal tended to decrease. This observation suggests that the concentration of Li metal within the TiN NT was most abundant near the LLZTO/TiN NT interface and progressively decreases in the direction towards the current collector during the charging process. In other words, the Li metal deposition occurs primarily at the electrochemically active LLZTO/TiN NT interface, and as the charging progresses, the continued precipitation spurs the Li metal to move along the capillary walls of the TiN NT via a diffusional creep mechanism. Consequently, the Li metal is fully deposited inside the tubules when the SOC reaches 100% (Fig. 2b). The cross-sectional scanning electron microscopy (SEM) image of the fully charged TiN NT also confirmed that Li metal was successfully deposited inside the tubular structure (Supplementary Fig. 8). To assess the impact of Li metal plating/stripping on the TiN NT wall structure, we conducted the cross-sectional SEM analysis of TiN NT, examining the diameter of the nanotubes before and after the cycling (Supplementary Fig. 9). The results indicate that the diameter of TiN NT remains unchanged even after the cycling process. This confirmed the robust mechanical strength of our TiN NT architecture, demonstrating its ability to withstand the stress associated with the Li plating/stripping process. These findings align with prior studies, showing that diffusional creep serves as a major lithiation mechanism in a MIEC tubular matrix featuring 100 nm wide and 10–100 μm deep channels in an SE-based

battery system[19,26]. Additionally, we conducted an extensive chemical analysis of the deposited Li inside the TiN NT capillaries using TOF-SIMS 2D mapping of the charged samples at various sputtering times (Fig. 2c). For the case of the 20% SOC sample, a strong Li⁺ signal was initially detected across the entire surface of the TiN NT. However, the signal gradually weakened and eventually vanished at a sputtering time of 5000 s, supporting that Li deposition starts at the LLZTO/TiN NT interface and creeps towards the current collector direction. In contrast, for the case of 100% SOC, a strong Li⁺ signal was detected throughout the sputtering time, indicating that Li metal was successfully deposited inside the TiN NT capillaries over the course of the charging process. These results suggest that the TiN NT anode can effectively accommodate Li metal inside its porous structure, mitigating volume expansion in the interlayer between the Ag-C and the current collector. This enables near-strain-free operation, as minimized volume change can prevent associated mechanical stress and deformation applied to the cell during the charging/discharging process. Furthermore, the cross-sectional focused ion beam (FIB)-SEM analysis of the charged cell was performed to detect any residual lithium metal at the interface (Supplementary Fig. 10a). The EDS elemental mapping images reveal distinct regions corresponding to TiN NT (Supplementary Fig. 10b, c) and Ag-C layers (Supplementary Fig. 10e, f), characterized by a homogeneous distribution of their component elements. Evidently, the presence of O element in both FIB-milled TiN NT and Ag-C regions indicates the existence of Li (in oxide form due to air exposure), which are expected to manifest as $Li_{bcc}$ deposition within TiN NT and AgLi alloy within the Ag-C, respectively (Supplementary Fig. 10d, g). This result accentuates that the amount of residual lithium metal at the interface is negligible and also highlights the effective $Li_{bcc}$ accommodating capability of TiN NT.

## Analysis of the Li propagation behaviour through the Ag-C interlayer and volume change suppression capability of the TiN NT architecture

Recent studies have shown that an Ag-C interlayer serves as an effective dendrite protection layer for both sulfide and oxide SEs by directing Li deposition towards the Ag-C/current collector interface[18,27]. However, compared to sulfide SE-based anode-free cells, the use of an Ag-C interlayer alone in oxide SE-based anode-free (or Li-free) cells has not yet produced satisfactory cyclability[17,34]. This limitation can be attributed to the brittle and inelastic nature of oxide SE, which makes it susceptible to interfacial disintegration and mechanical failure from recurring local stress and volume changes during cycling[35–37]. Nonetheless, the incorporation of Ag-C has proven instrumental in our system, as its role extends beyond guiding the deposition of reduced Li towards the negative electrode. It also plays a crucial role in achieving uniform interfacial contact at the LLZTO/TiN NT interface. This is particularly evident when TiN NT is used alone in the full cell without an Ag-C interlayer. As illustrated in the voltage profile result (Supplementary Fig. 11), the cell experienced an immediate short-circuit during the charging process and failed to operate thereafter. Thus, accommodating reduced Li within the free volume space of the TiN NTs necessitates the use of an Ag-C interlayer to establish seamless interfacial contact at the LLZTO/TiN NT interface. To effectively mitigate the volume changes in oxide SE-based AFSSBs during cell operation, it is imperative to closely examine the alterations occurring at the interface and the bulk components of the cell during the charging/discharging process. In this regard, *operando* analysis using optical microscopy (OM) was conducted to observe the Li propagation behaviour through the Ag-C interlayer during the charge/discharge process, while cross-section SEM was employed to examine the volume change of the Ag-C interlayer. Furthermore, the volume change suppression capability of the TiN NTs was examined by measuring the variations in the thickness of the full cell during cycling.

To conduct *operando* OM analysis, we utilized a specially designed cell consisting of a Cu mesh (opening size: 300 μm × 300 μm) current collector to monitor the Li evolution in the Ag-C interlayer during the charging/discharging process (Fig. 3a and Supplementary Fig. 12a). To improve the cell's cyclability, we employed a new pre-lithiation technique that involves attaching a halo Li foil onto the Ag-C coating layer via CIP under 250 MPa. Due to the facile alloying property of Ag to form Ag-Li, partial physical contact between Ag-C and Li was sufficient for pre-lithiation to occur[38,39]. As evidence, the cell with the pre-lithiated Ag-C (Pre_Li_Ag-C) interlayer exhibited superior Li plating and stripping performance compared to that with the pure Ag-C interlayer (Supplementary Fig. 13), along with enhanced specific capacity and reversibility in the first cycle. Using this setup, we observed the Li evolution from the Ag-C interlayer under an applied galvanostatic current of 1.0 mA cm⁻² at 25 °C. A real-time observation of the Ag-C layer during the charging process (Fig. 3b) shows that island-type Li precipitates can be found on the Ag-C layer after 60 min, and they grow in size and number after 120 min of deposition (Fig. 3c). As more clearly shown in Supplementary Video 1, Li begins to protrude from the surface of the Ag-C layer in the form of interspersed islands and continues to grow over time during charging. It then develops into filamentary Li as evidenced by the out-of-focus features in the OM image caused by the elevation of height above the Ag-C surface. Moreover, a closer examination of the Ag-C surface via SEM analysis reveals a stream of Li metal protruding adjacent to the Cu mesh (Supplementary Fig. 12b). Cross-sectional FIB-SEM and EDS image at this border disclosed a layer of deposited Li metal between the Cu mesh and Ag-C (Supplementary Fig. 12c, d). During the subsequent discharging process, the Li deposits on the Ag-C layer gradually recede back into the Ag-C layer, but some remaining Li islands can be observed at the end, revealing incomplete stripping of Li at the interface. To further examine whether the lithiation process accompanies a volume change in the Ag-C interlayer, a similar cell, but with Li metal as an anode, was cycled at 0.1 C and 0.5 C, respectively. As shown in the cross-sectional SEM images (Fig. 3d), there was no discernible change in the thickness of the Ag-C layer, even when the cell was lithiated under a higher current density (0.5 C). Similarly, there was a negligible difference in the thickness of the deposited Li layer between the 0.1 C and 0.5 C lithiated cells (Supplementary Fig. 14), indicating that the thickness of the Ag-C layer can be well maintained under a charging rate of 0.5 C (1.6 mA cm⁻²). These results thus validate that the Ag-C interlayer is suitable for use in conjunction with TiN NTs.

Based on the above observation, we investigated the volume changes of the TiN NT-incorporated full cell during cycling via a 4-point laser displacement sensor (Fig. 4a). To enable direct measurement from the current collector surface, we assembled a pouch-type full cell with a central hole that was sealed with a Mylar film to allow the laser beam to pass through. Additionally, the cell was vacuum-sealed to prevent any external interference and paired with a potentiostat for cycling control. The internal pressure was generated solely by the vacuum inside the pouch (-1 bar), with no external stacking pressure applied (Supplementary Fig. 15). Using this setup, we compared three pouch-type full cells with different interlayer/anode configurations: Ag-C/Li, Pre_Li_Ag-C/Cu, and Pre_Li_Ag-C/TiN NT. We first examined the change in thickness of each full cell under the same charging current density of 0.64 mA cm⁻² (Fig. 4b). The results in Fig. 4c demonstrate that the pouch cell with the Pre_Li_Ag-C/TiN NT exhibited approximately 85% suppression in Li deposition-induced volume expansion compared to the other two cells by the end of the charging time. Moreover, the change in thickness of the pouch cells from their initial state was measured as a function of cycle number (Fig. 4d). The pouch cell with Ag-C/Li significantly increased in thickness during cycling, leading to a short circuit after the 7th cycle, which might be attributable to interfacial disintegration originating from the volume change during cycling. The pouch cell with the Pre_Li_Ag-C/Cu

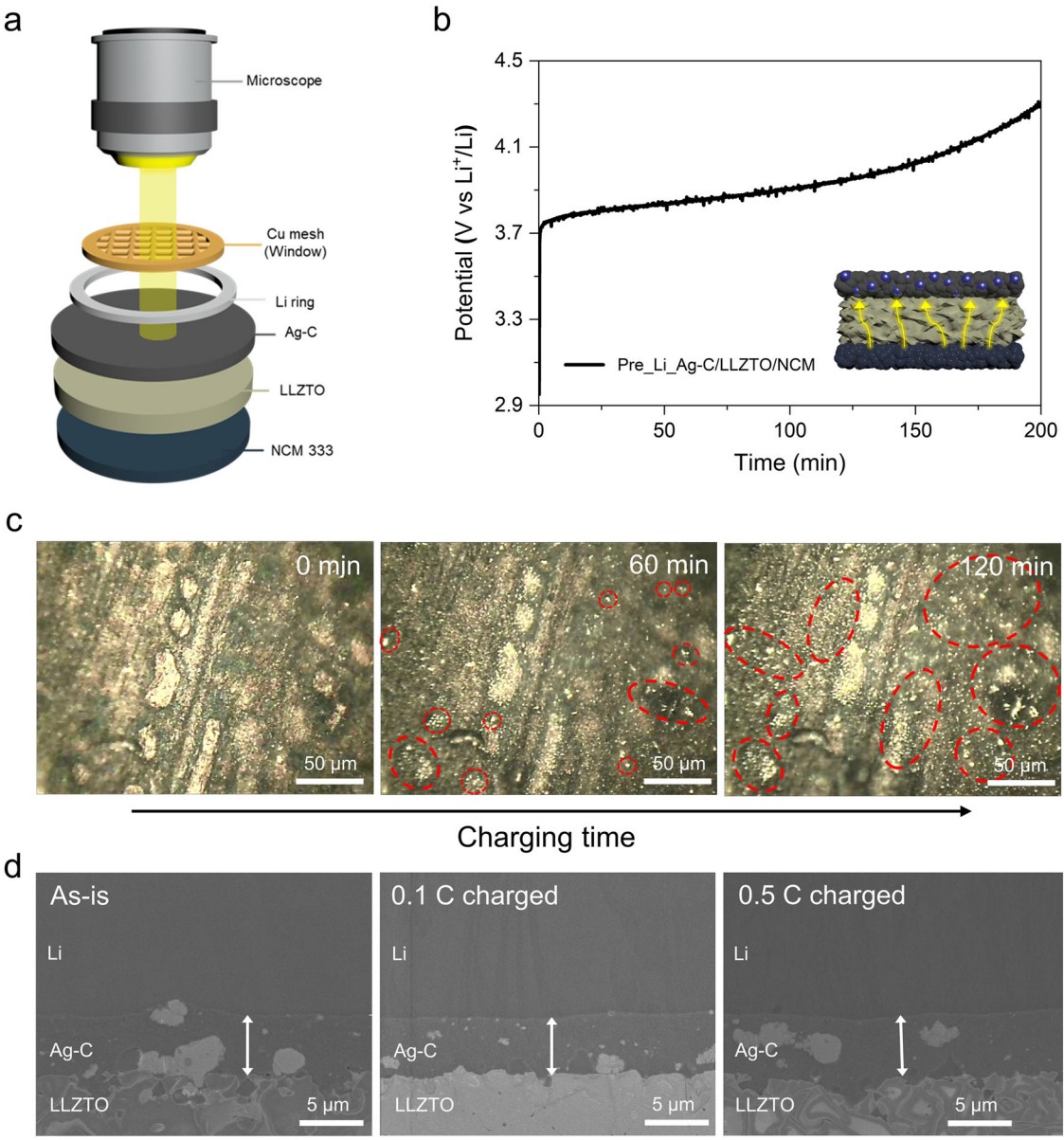

**Fig. 3 | Analysis of the Li propagation behaviour and volume change of the Ag-C interlayer. a** Schematic of the in-situ optical microscopy (OM) setup to observe Li propagation through the Ag-C interlayer. **b** Galvanostatic charging voltage profile of the full cell from in situ OM analysis with the inset showing the direction of Li propagation. **c** In situ OM images of the Ag-C layer during the galvanostatic deposition of Li. The formation of Li precipitates intensifies as the deposition time increases (red circle). **d** Cross-sectional SEM images of the full cell before and after charging at current densities of 0.1 C and 0.5 C.

experienced a rapid thickness increase in the first cycle but reached a plateau afterwards with a consistent increment from the initial state. In contrast, the pouch cell with the Pre_Li_Ag-C/TiN NT decreased in thickness after the first cycle, presumably due to the intratubular diffusion process, and then maintained its thickness with only minute variation throughout the measuring cycles. Notably, the Pre_Li_Ag-C/TiN NT-incorporated pouch cell exhibited an average change in thickness of 3.5 μm from its pristine state, significantly lower than that of the Ag-C/Li (42.1 μm) and Pre_Li_Ag-C/Cu (15.1 μm) installed pouch cells (Supplementary Table 1). Additionally, when measuring the change in thickness during the lithiation/delithiation process (Supplementary Fig. 16), the Pre_Li_Ag-C/TiN NT-installed pouch cell displayed the lowest average change in thickness (4.3 μm) compared to Ag-C/Li (16.3 μm) and Pre_Li_Ag-C/Cu (11.6 μm) (Supplementary Table 1). Furthermore, the comparison of capacity retentions among the pouch cells from the above test (Fig. 4e) highlights the outstanding performance of Pre_Li_Ag-C/TiN NT compared to the other two cells.

Interestingly, the capacity of the Pre_Li_Ag-C/TiN NT mounted pouch cell increased consistently until the 7th cycle, presumably due to the emergence of the previously deposited Li in the nanotubes in the subsequent discharging step. It has been reported that the creeping of Li into the nanotubes can hermetically seal the structure, resulting in an internal pressure difference that might lead to heterogeneity in Li filling. Consequently, these analyses unequivocally demonstrate the superior volume/strain buffering capability of TiN NT, underscoring its effective $Li_{bcc}$ accommodation within the nanotube arrays. It is also worth noting that the TiN NT incorporated full cell could be cycled without incurring internal short circuits or capacity fading, even under near-zero stack pressure conditions. Therefore, minimizing the thickness change is a crucial factor for enabling the cell to operate effectively under low stack pressure conditions.

To further investigate the volume change buffering capability and electrochemical performance of TiN NT during harsh lithium deposition/stripping conditions, we fabricated two symmetric cells, each

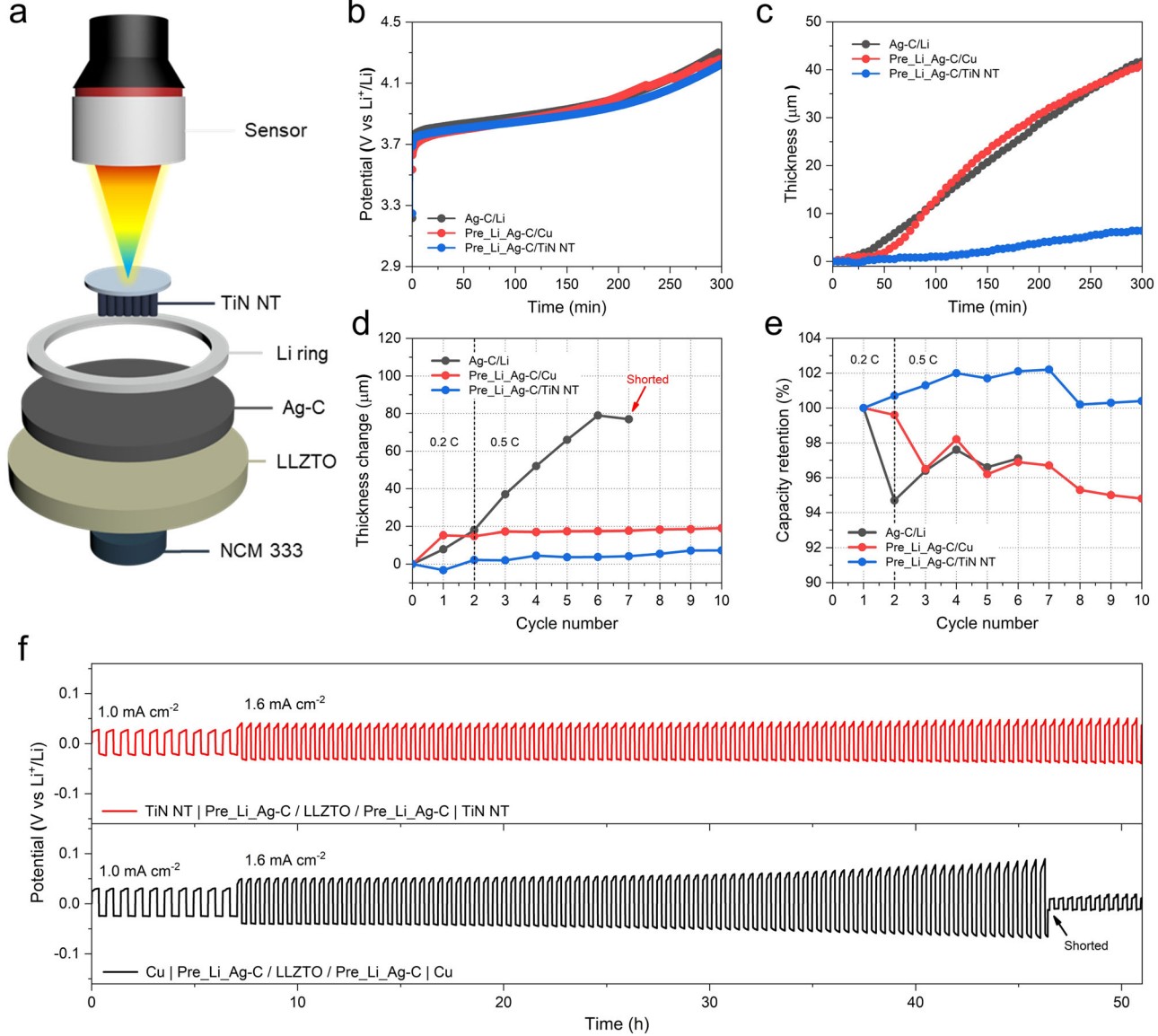

**Fig. 4 | Volume change suppression capability of the TiN NT anode architecture. a** Schematic of the volume change measurement setup comprised of the laser displacement sensor and a pouch-type full cell. **b-c** Voltage profiles (**b**) and thickness changes (**c**) of various pouch-type full cells as a function of charging time (min). **d-e** Thickness changes (**d**) and capacity retentions (**e**) of various pouch-type full cells from their initial state as a function of cycle number. **f** Comparison of the galvanostatic cycling performance of the TiN NT|Pre_Li_Ag-C/LLZTO/Pre_Li_Ag-C|TiN NT and Cu|Pre_Li_Ag-C/LLZTO/Pre_Li_Ag-C|Cu symmetric cells at 25 °C at current densities of 1.0 mA cm$^{-2}$ and 1.6 mA cm$^{-2}$.

containing TiN NT and Cu attached to opposite sides of the Pre_Li_Ag-C interlayer double-affixed LLZTO (hereafter referred to as TiN NT and Cu symmetric cells). Subsequently, we measured their thickness change during the initial charging process (0.2 C) using a laser displacement measurement setup (Supplementary Fig. 17a). The results show that by the end of the charging process, the thickness of Cu symmetric cell reached 20 μm, whereas the TiN NT symmetric cell only reached 12 μm, which is nearly half of the Cu symmetric cell (Supplementary Fig. 17b). This validates the efficient volume buffering capability of TiN NT. Additionally, Li plating/stripping tests were carried out under varying current densities at 25 °C. The result in Fig. 4f demonstrates that the Cu symmetric cell undergoes a continuous increase in polarization at a current density of 1.6 mA cm$^{-2}$ and ultimately succumbs to a short circuit in less than 40 h. In contrast, the TiN NT symmetric cell delivers stable stripping/deposition cycles without noticeable overpotential, indicating an improvement in the critical current density. Moreover, the electrochemical impedance

spectroscopy (EIS) measurement of the same symmetric cells before and after the cycle was conducted to evaluate their change in the cell impedance (Supplementary Fig. 18a, b). The Nyquist plot and corresponding fitting result (Supplementary Table 2) revealed that the TiN NT symmetric cell exhibits much lower interfacial resistance compared to its Cu counterpart before the cycle (3.96 vs. 5.76 Ω cm$^2$) and the trend persists after the cycle (5.56 vs. 9.03 Ω cm$^2$), signifying the efficacy of the TiN NT in maintaining the interfacial contact during cell operation.

## Electrochemical performance of the TiN NT-incorporated AFSSB full cell

To further validate the near-strain-free feature of the TiN NTs, electrochemical performance tests were conducted using the AFSSB full cell configuration of the TiN NT|Pre_Li_Ag-C/LLZTO|NCM333. The NCM333 cathode (3.2 mAh cm$^{-2}$) was wetted with an IL-based electrolyte solution to reduce the interfacial impedance between the

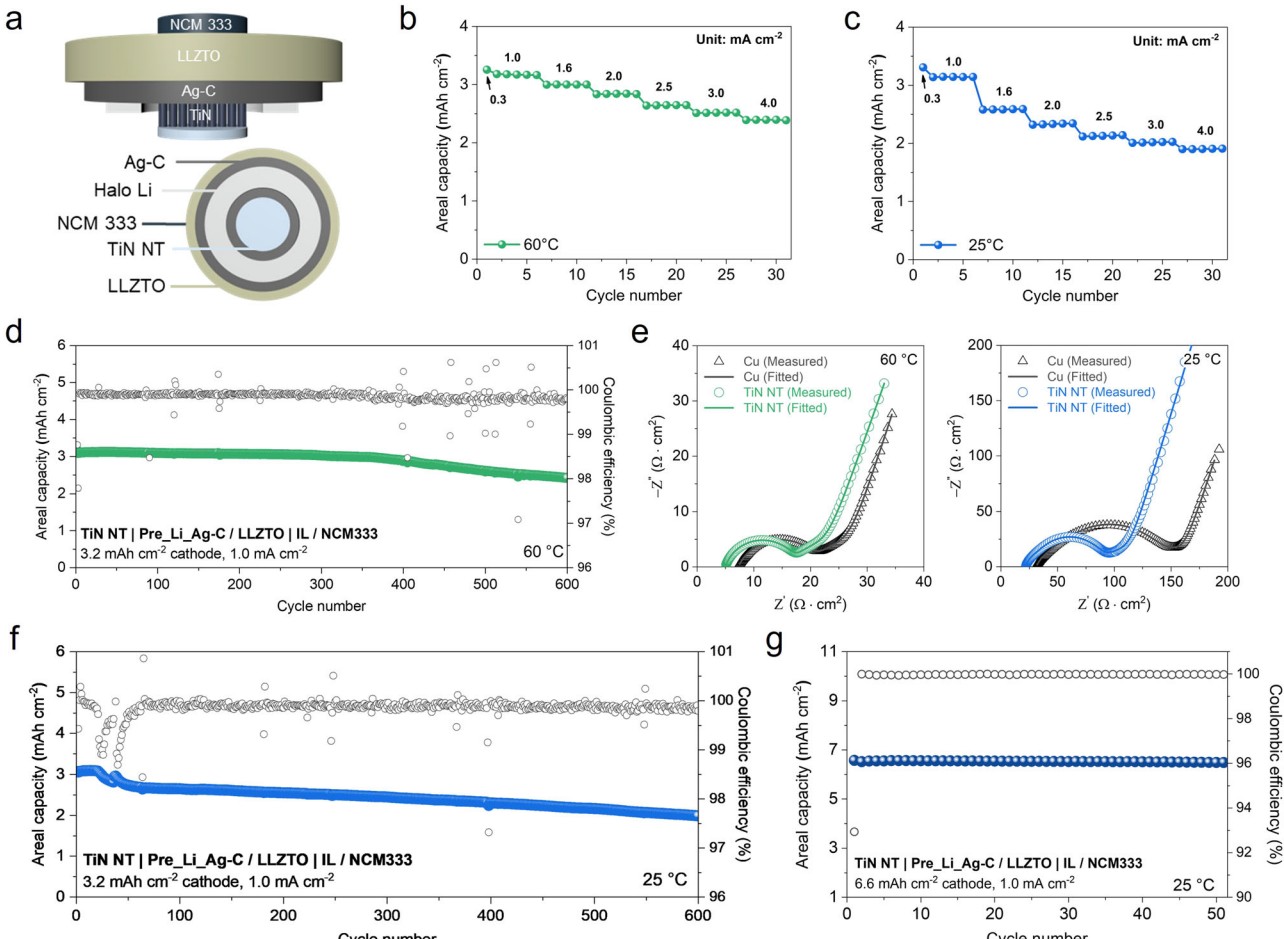

**Fig. 5 | Electrochemical performance of the TiN NT-incorporated garnet-type SE-based AFSSBs. a** Schematic of the assembled AFSSB full cell with a near-strain-free TiN NT anode architecture. Rate capability of the AFSSB full cell at **b** 60 °C and **c** 25 °C. **d** Long-term cycling performance of the AFSSB full cell at 60 °C. **e** Nyquist plots of AC-impedance spectra obtained from the AFSSB full cells with different anodes (Cu vs. TiN NT) at 60 °C (left) and 25 °C (right). **f** Long-term cycling performance of the AFSSB full cell at 25 °C. **g** Cycling performance of the AFSSB full cell with a high cathode capacity of 6.6 mAh cm$^{-2}$ at 25 °C.

LLZTO and the positive electrode. The schematic of the fabricated full cell is illustrated in Fig. 5a. To evaluate the rate performance of the full cell, a rate capability test was conducted with varying current densities, ranging from 0.3 mA cm$^{-2}$ to 4.0 mA cm$^{-2}$, under two distinct temperature conditions: 60 °C and 25 °C (Fig. 5b, c). The utilization of different temperature settings aimed to examine the impact of temperature on the diffusion creep rate of Li metal inside the nanotube confinement. As shown in the results, both full cells delivered a stable discharge capacity even at a high current density of 4.0 mA cm$^{-2}$. However, the rate capability of the full cell at 60 °C consistently outperformed that at 25 °C across all current density conditions (Supplementary Fig. 19). This improved performance at elevated temperatures can be attributed to various factors, including the accelerated Li metal creep rate within the nanotube due to the enhanced kinetics and the concurrent effect of an increase in the ionic conductivity of LLZTO. Moreover, the long-term cycling performance was evaluated by cycling the full cell consisting of the same NCM333 cathode (with an areal capacity of 3.2 mAh cm$^{-2}$) under a current density of 1.0 mA cm$^{-2}$ at 60 °C (Fig. 5d). The cell delivers an initial areal capacity of 3.15 mAh cm$^{-2}$ and a discharge capacity retention of 78.3% after 600 cycles with a coulombic efficiency of 99.8%. Additionally, we conducted EIS analysis on TiN NT and Cu installed full cells at both 25 and 60 °C to comparatively examine their impact on cell impedance under different temperature conditions. Figure 5e and Supplementary Fig. 20 show the Nyquist plots of the impedance spectra consisting of two depressed arcs, which overlap in the high-frequency range,

Warburg impedance, and a capacitive line (blocking region) in the low-frequency range. The equivalent circuit was implemented to quantitatively analyze the experimental impedance spectra, and the complex nonlinear least-squares (CNLS) fitting method was used to determine the bulk ionic resistance ($R_b$) and interfacial resistance ($R_{ct} = R_1 + R_2$) (the values are listed in Supplementary Table 3)[18,40,41]. Notably, a more pronounced difference in $R_b$ and $R_{ct}$ values between the TiN NT and Cu full cells occurs at 25 °C (than at 60 °C) due to the overall increase in bulk and interfacial resistance associated with the lower ionic conductivity of the LLZTO SE at room temperature. Amidst of this limited ionic conduction, the impedance contribution at the Ag-C and the current collector interface becomes dominant. This is corroborated by the significantly lower $R_b$ and $R_{ct}$ values (22.0 and 62.1 Ω cm$^2$) of the TiN NT full cell compared to its Cu counterpart (31.1 and 107.4 Ω cm$^2$), which can be ascribed to superior charge-transfer kinetics and interfacial stability at the Ag-C|TiN NT interface. Evidently, the long-term cycle performance of the TiN NT-incorporated full cell at room temperature (25 °C) under a current density of 1 mA cm$^{-2}$ showed remarkable cyclability over 600 cycles without significant capacity degradation (Fig. 5f), and the cell recorded an outstanding average coulombic efficiency of 99.8% for the entire cycle. In contrast, the full cell with the Cu current collector experienced a noticeable capacity deterioration after 20 cycles when operated under the same conditions (Supplementary Fig. 21). This implies a highly reversible Li plating/stripping process enabled by the enhanced interfacial stability and volume-buffering capability of the TiN NTs. Furthermore, even when

using a cathode with a significantly higher capacity (6.6 mAh cm$^{-2}$), the cell could deliver a stable cycling performance over 50 cycles with an initial capacity of 6.56 mAh cm$^{-2}$ and a capacity retention of 98.6% after 50 cycles (Fig. 5 g). The cell also maintained an average coulombic efficiency of 99.9% throughout the cycles. As summarized in Supplementary Table 4, our TiN NT-incorporated AFSSBs were competitive in cycling performance and current density compared to previously reported garnet-type SE-based SSBs with various cathode active materials.

## Discussion

In conclusion, we presented the concept of utilizing a diffusional creep mechanism to develop a highly stable and long cycling garnet-type SE-based AFSSB full cell. The introduction of an MIEC material-based TiN NT as an anode interface enabled reversible accommodation and diffusion of reduced Li within the nanotube arrays via interfacial-diffusional creep during the charging/discharging process. We also addressed the commonly overlooked issues of internal strain and volume expansion caused by the repeated Li plating/stripping process in AFSSBs. From *operando* OM and cross-sectional SEM analysis, we confirmed that the Ag-C interlayer can direct Li deposition at the interlayer/negative anode interface while maintaining its thickness even under high current density conditions, indicating its crucial role in both dendrite protection and volume stability. Therefore, by regulating the Li transport through the Ag-C interlayer and accommodating the accrued Li deposits within the TiN NT nanotube structure, we constructed an interlayer/interface design with exceptional volume suppression capability during lithiation/delithiation. Given these synergistic properties of Ag-C and the TiN NTs, we fabricated an AFSSB full cell, which, compared to the Cu current collector, exhibited an almost tenfold suppression in volume expansion and delivered remarkable cycling performance at room temperature (25 °C) over 600 cycles with an average coulombic efficiency of 99.8%. The successful implementation of the diffusion creep mechanism in this study is expected to provide a promising avenue for the development of high-performance garnet-type SE-based AFSSB full cells.

## Methods

### Fabrication of the TiN NT anode architecture

The TiN NT anode architecture was prepared by a two-step anodization process. First, the Ti foil (0.25 mm, 99%, Alfa Aesar) was sonicated in a deionized (DI) water-ethanol-acetone mixed solution. The anodization process was carried out in two-electrode configurations with a Pt foil counter electrode. The first-step anodization process was performed at 60 V in a solution of 0.5 wt% NH$_4$F (≥98%, Sigma-Aldrich) in ethylene glycol (anhydrous, 99.8%, Sigma-Aldrich) with 2 vol% deionized (DI) water for 1 hour. After the first-step anodization, the as-grown nanotube array of Ti foil was removed by sonicating in DI water. The second-step anodization process was also performed with the same procedure on anodized Ti foil. After the process was completed, the TiO$_2$ nanotube array was washed with ethanol to remove the residual aggregate and was annealed at 450 °C for 1 hour under air conditions. Subsequently, the annealed TiO$_2$ nanotube array was annealed in a tube furnace at 800 °C for 1 hour under ammonia (99.999%) conditions. Finally, the TiN NT anode architecture was obtained.

### Preparation of LLZTO solid electrolyte pellets

LLZTO (Li$_{6.4}$La$_3$Zr$_{1.7}$Ta$_{0.3}$O$_{12}$) powder was synthesized from a precursor mixture of Li$_2$CO$_3$ (>99.0%, ChemPoint), La$_2$O$_3$ (98.6%, Moly-Corp), Ta$_2$O$_5$ (99.99%, Sigma-Aldrich), and ZrO$_2$ (98%, Zircoa Inc.) using a solid-state reaction method. The mixed powder was calcined in air at 950 °C for 5 h followed by 1200 °C for 5 h to yield LLZTO powder. The calcined powder was then ball-milled in air with zirconia balls for 10 min at 300 rpm using a planetary mill (Pulverisette 7,

Fritsch, Germany). The ball milling was repeated 12 times at 5-min intervals.

To fabricate a dense pellet, LLZTO powder (100 g) was hot-pressed in a graphite die at 3 kpsi, followed by sintering at 1100 °C for 2 h in an Ar atmosphere at a heating rate of 300 °C/h. The relative density of the pellet was estimated to be >98% with respect to the theoretical density of LLZTO calculated from the X-ray diffraction (XRD) data. The pellet was cut into with (14 mm diameter, 360 μm thickness) using a laser cutter in air. Next, it was subjected to ultrasonic cleaning in hexane for 10 min and heat-treated at 800 °C for 1 h in dry air. Finally, the pellet surface was polished to a thickness of ~350 μm using polishing machines (LaboForce-3, Struers).

Acid treatment was performed to clean the LLZTO surface by simply immersing the discs into a glass bottle with 1 M HCl solution (in distilled water) in a dry room (dew point, −60 °C) at 25 °C for 20 min at a weight ratio of 1:10 (pellet: acid solution). To prevent the local variation in concentration in the acid solution due to the released Li or prevent the close contact of the electrolyte to the glass container, the container was rolled in the bottle-roller at approximately 60 rpm during protonation. We then removed the solution, washed the discs with ethanol, and dried them in a dry room.

### Cell assembly and electrochemical characterizations

We employed a quasi-solid-state cell. In each quasi-solid-state cell, an ionic liquid, and a solid oxide electrolyte (LLZTO) were used as the cathode and anode electrolytes, respectively. An Ag-C layer coated on a 10-μm-thick stainless steel (SUS) foil was prepared using a mixture of carbon black powder (99.7%, average particle size = 38 nm, Asahi carbon) and Ag nanoparticles (D50 = 60 nm)[27]. Ag and carbon black powder were mixed in a weight ratio of 1:3 in N-methylpyrrolidone (Sigma-Aldrich) with 7 wt% polyvinylidene fluoride (Solvay) as a binder under constant stirring (1000 rpm) for 30 min using a mixer (Thinky Corporation, AR-100). The resulting slurry was then coated on SUS foil using a screen printer and dried in air at 80 °C for 20 min. The coated Ag-C layer was further dried under a vacuum at 100 °C for 12 h. The Ag-C layer was attached as an anode interlayer on the acid-treated LLZTO surface via cold-isostatic pressing (CIP) under 250 MPa. After the SUS foil was peeled, a TiN NT or Cu foil was attached to the Ag-C surface via 250 MPa CIP.

Commercially available (Li$_{1+x}$(Ni$_{0.33}$Co$_{0.33}$Mn$_{0.33}$)$_{1-x}$O$_2$ (NCM333, loading capacity: 3.2 g/cc, active material: 96 wt%; thickness: 50 μm; Samsung SDI) was employed as a cathode and as a current collector, Al foil (9 μm foil, Nippon Foil Mfg. Co., LTD) was used as received. A N-methyl-N-propyl pyrrolidinium bis(fluorosulfonyl) imide (Pyr13FSI, 99.9%, water content <20 ppm, Kanto Chemical Co. Inc.) ionic liquid mixed with a 2 M lithium bis(fluorosulfonyl)imide (LiFSI, 99.9%, water content <10 ppm) salt was used as the catholyte. The catholyte solution (20 wt% relative to the cathode) was infiltrated into the cathode in a dry room (dew point, −60 °C), followed by maintaining a vacuum state for 2 h. When the residual solution on the cathode surface was removed using Kimwipes, the solution uptake by the cathode was ~7 wt %. In our cell configuration, the ionic liquid comprises ~0.055 wt% of the entire electrolyte. We placed the ionic-liquid-infiltrated cathode on the cathode side of the LLZTO. For Ag-C pre-lithiation, a halo Li was prepared by punching the Li-Cu foil (with a Li thickness of 20 μm) into a ring with exterior and interior diameters of 12 and 8 mm, respectively. Subsequently, the prepared halo Li was affixed to the Ag-C coated side of the LLZTO via 250 MPa CIP for 3 min, followed by incubation in an oven set at 45 °C for 7-12 h. The entire cell components were then assembled in a CR2032 coin cell, including a spring and two 0.5 T disks (as shown in Supplementary Fig. 15a), resulting in an internal pressure of approximately 0.6 MPa after clamping.

Potentiostatic electrochemical impedance spectroscopy (PEIS) measurements were conducted at 25 °C with an open circuit in the potentiostatic mode over a frequency range from 0.1 Hz to 10 kHz using

an alternating current (AC) perturbation of 10 mV with a frequency response analyser (Solartron, SI 1255 FRA) in conjunction with a potentiostat (Solartron, SI 1287 ECI). All PEIS data were recorded with 10 frequencies per decade. Before conducting the PEIS measurements, the cells were kept under open-circuit potential for 10 min. A battery cycler (Toscat-3100, Toyo System) was employed to measure the charge–discharge curves of the quasi-solid-state cells at 25 °C. The cells were cycled with a constant current (CC)−constant voltage (CV) charging and CV discharging mode in the potential ranges of 2.8–4.3 (vs. Li/Li$^+$) for NCM333. We evaluated five cells of each sample to ensure data reliability.

## Characterization methods

The morphology and X-ray diffraction (XRD) patterns of the TiN NTs were obtained by field emission scanning electron microscopy (JEOL-7800F) and X-ray diffraction analysis (Ultima IV, Rigaku), respectively. Morphological and chemical changes as a function of cell operation were analysed by processing cross-sections of the cells. The cross-sectional samples were prepared with a cross-section polisher (IB-19520CCP, JEOL Ltd.) cooled using liquid nitrogen to reduce surface damage from the high-energy Ar-ion beam. The morphological and chemical changes caused by the Li reaction were analyzed using a scanning electron microscope (SU-8030, Hitachi) and energy-dispersive X-ray spectroscope (X-max 80, Oxford). To prevent atmospheric exposure of the samples, the samples were transferred to the electron microscope in a transfer vessel inside a glove box, and the samples were loaded in the vacuum environment of the electron microscope. The Li deposition behaviour was investigated using time-of-flight secondary ion mass spectrometry (TOF-SIMS 5, IONTOF) with a 30 keV Bi$_1^+$ ion beam. Oxygen (2 keV) was used for sputtering the Li-deposited TiN NT samples.

## In-situ optical microscope (OM) analysis and cell thickness measurement

In the 2032-coin cell, an 8 mm diameter hole was made on the coin cell case to observe morphological changes in the Ag-C layer during cell operation. The 2032-coin cell was assembled inside a dry room with Cu mesh on the Ag-C layer, LLZTO solid electrolyte (Toshima), and NCM333 cathode. The in-situ cell was placed into a printed circuit board (PCB) coin cell holder, and then it was fixed on the sample stage of an optical microscope (Micro support, Axis pro) inside a glove box. A PCB holder was connected to a potentiostat (Bio-Logics, Bistat) for electrochemical cycling.

As shown in Supplementary Fig. 15, the in-situ cell thickness measurement setup comprises a sensor head (CL-L030, KEYENCE), an optical unit (CL-P030N, KEYENCE), and an ultracompact switch-mode power supply (CA-U4). This configuration was interconnected with a constant temperature oven (Espec, Temperature chamber SU-222 Bench top type) and potentiostat (Solartron 1287) to enable simultaneous charging/discharging, and thickness measurement at a constant temperature. The cell used for in-situ OM analysis followed a conventional coin-cell structure, except for the use of a spring to apply the pressure. Instead, it was vacuum sealed in an aluminum pouch, with extended line for electrical connection (as shown in Supplementary Fig. 15b). With this configuration, we could prevent air exposure and apply a constant atmospheric pressure (-1 bar) by maintaining a vacuum state.

## Data availability

The authors declare that the data supporting the findings of this study are available within the article and its Supplementary information. Additional data are available from the corresponding author upon request.

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

## Acknowledgements

This work was supported by Samsung Advanced Institute of Technology. The National Research Foundation of Korea (NRF) grant funded by Ministry of Science and ICT (RS-2023-00302697 (J.H.P.), NRF-2022M3J1A1085397 (Phased development of carbon neutral technologies) (J.H.P.)) also support this work.

## Author contributions

J.-S.K and J.H.P. supervised the project. K.H.K., M.-J.L. and M.R. conceived the idea, designed the experiments, and co-wrote the paper. T.-K.L. and J.H.L. assisted material synthesis and conducted the characterizations. C.J. performed the In-situ optical microscope and thickness measurement. All the authors discussed the results and commented on the manuscript.

## Competing interests

The authors declare no competing interests.
