## [Peer Review File · Nature Communications]

Near-strain-free anode architecture enabled by interfacial diffusion creep for initial-anode-free quasi-solid-state batteriesREVIEWER COMMENTS

Reviewer #1 (Remarks to the Author):

In this work, the authors employ the titanium nitrate nanotube structure combined with an Ag-C interlayer to address the anisotropic stress arising from repetitive Li deposition layers during cycling. By incorporating an MIEC material into the anode interface, the author emphasizes the enhanced movement of Li within the nanotube arrays, mainly due to diffusional creep during charge/discharge cycles. Remarkably, the battery demonstrated excellent performance. The article is well-constructed, possesses a unique perspective and provides new insight and methodologies. However, certain expressions and explanations are not very clear. In my view, several critical questions must be addressed before proceeding further. The identified issues are as follows:

1. The authors claim that the effect of deposition behavior of lithium metal and suppressing volume expansion occurs via the interfacial-diffusional Coble creep mechanism. How does the author prove that lithium metal behaves as Coble creep mechanism rather than Nabarro-Herring creep in MIEC? In addition, the rate of diffusion creep is greatly affected by temperature. The authors should include a discussion of diffusion creep when studying electrochemical performance at different temperatures (Figure 5).
2. In Figures 3a-c, a new experimental technique was employed to study the Li propagation through the Ag-C interlayer. However, the data presented seems insufficient to provide a full understanding of the Li propagation behavior. Could the authors present evidence showing a reduction in residual lithium metal after introducing MIEC using this method? Additionally, could the presence of the copper mesh obscure some OM details because of the lithium metal deposited on it? Furthermore, there is an inconsistency between the "Ni mesh" label in Fig 3a and the mention of "Cu mesh" in the text.
3. Figure 2 presents the lithium metal deposition process. Would the author consider adding pertinent ion fragmentation data for TiN to further substantiate this experiment?
4. The authors should explain more specifically how Fig 4f combine the symmetric cells with "volume change buffering capability". The current rationale is not persuasive.
5. The authors' impedance analysis seems oversimplified. Given the presence of multiple phases in the system, there might be numerous interfaces and associated electrochemical process responses.

Introducing temperature as a variable further complicates the analysis. Thus, relying solely on the comparison of "the size of the semicircle" appears inadequate. It would be beneficial if the fitting circuits for Fig 5e and s13 were provided. Highlighting the relevant electrochemical processes and subsequently drawing a detailed comparison would make the discussion more comprehensive.

Reviewer #2 (Remarks to the Author):

This manuscript describes a 3-D mixed conducting host structure in conjunction with a Ag-C interlayer to manage Li plating and stripping in garnet-based solid-state batteries. The results are a significant advance over the state-of-the-art and are well-presented. Therefore, I am happy to recommend publication after the following comments are addressed.

1. Operando should be italicized when used
2. What is the internal pressure of the cells used? How does it change during cycling? Could these changes impact performance, and how would they be different depending on form factor of the cells? Reducing stack pressure requirements is key to SSB development, so it should be addressed here. Does this approach to minimizing thickness change also change pressure requirements?
3. How much excess Li remains in the Li halo used for pre-lithiation? Is this Li halo active during cycling? If used in practice, how much would this excess Li impact energy density (volumetric and gravimetric?)
4. What are the sources of degradation during cycling in these full cells? Why is impedance increasing over time?

Reviewer #3 (Remarks to the Author):

In this work, the authors proposed an MIEC-based TiN nanotube structure capable of accommodating lithium through diffusional creep, along with an Ag-C interlayer. In an anode-free system, repeated Li plating and stripping cause significant volume changes, posing critical problems when using oxide solid electrolytes. The authors addressed this strain issue by utilizing TiN nanotubes as a 3D host for strain-free anode architecture. Through this strategy, remarkable cyclability was achieved. The in-situ analyses are interesting, however, the author's claims on the role of TiN NT and Ag-C interlayer were not clearly supported in this manuscript. The improvement in performance could be merely from the use of ionic liquid in the cathode and excess-Li in so-called

the pre-lithiation anode. I thereby recommend rejection of this manuscript. Questions and comments are provided below.

1. The authors claimed the effect of Ag-C interlayer as "The use of the Ag-C interlayer promoted the stable plating of Li metal within the TiN NT cavities and ensured continuous interfacial contact between the SE and TiN NT." (page 6, line 92-94), and the Li deposition in Ag-C layer and inside the TiN structure was analyzed using in-situ OM and TOF-SIMS. However, it is hard to elucidate the effect of Ag-C in this study. To clarify the effect of Ag-C for Li deposition, it is necessary to compare the performance of TiN NT without Ag-C. The improvement in cycling can be just ascribed to the effect of minimizing volume changes.

A. The full cell cycling of only TiN NT and Ag-C with TiN NT should be compared.

B. The Li deposition inside TiN NT for only TiN NT and Ag-C with TiN NT should be examined for comparison.

2. In relation to question #1, the author also stated "Additionally, a Ag-C interlayer is embedded between the LLZTO and TiN NT architecture to facilitate uniform Li deposition across the nanotubes by channelling the redox reaction at the interface between the Ag-C interlayer and TiN NT²⁷⁻²⁹." (page 4, line 67-70). Please clarify this statement. If this statement indicates that the redox reaction occurs at the interface between Ag-C and TiN or in Ag-C and TiN, this statement should be corrected. The redox reaction does not occur at the interface between Ag-C interlayer and TiN NT, but at the surface of LLZTO (or at the interface between Ag-C interlayer and LLZTO). The reduced Li metal move toward the current collector side by diffusional creep.

3. Using TiN NT imposes limitation on the achievable capacity. Currently, the diameter of the inner tubes in the structure is 100 nm. Using a larger diameter structure and/or porosity by thinning the tube wall could potentially increase lithium storage capacity.

A. Is it possible to control the diameter or wall thickness? Have you tried before?

B. Furthermore, within the current evaluation conditions, does the TiN NT layer expand in diameter? Is there any interfacial Li deposition at the interface between Ag-C and TiN or on the current collector? The cross-sectional images after Li deposition might help.

4. In ToF-SIMS data, the amount of Li within TiN NT progressively decrease toward the current collector side. This indicate that the TiN NT is not fully filled with Li. What is the maximum capacity when TiN NT is completely filled? Is there any interfacial deposition between Ag-C and TiN NT layer when you increase the capacity before TiN NT is completely filled? Then, the diffusional creep through TiN should be enhanced for fully accommodating Li inside TiN NT. This point with the related data should be addressed in the manuscript.

5. The Pre_Li_Ag-C is formed by simple compression using CIP with halo Li foil, as mentioned in page 9, line 155-156. I recommend adding information about this prelithiation process (such as amount of reacted lithium foil and reaction time) to the methods section.

A. Does the halo Li foil used in this prelithiation reaction participate entirely in the reaction, leaving no residual lithium layer?

B. In Fig. 3d, the cross-sectional SEM image of "As-is" shows a significant amount of lithium layers with a thickness of over 10 μm before cycling. What is the role of this lithium layer at this stage? Why does it exist? If extra Li layer exists during cycling, it should be not mentioned as "pre-lithiated". It is not a Li-free or anode-free system anymore.

C. What are the differences between Ag-C/Li and Pre_Li_Ag-C as compared in Fig. 4?

6. In Fig. 5e, the authors compared Cu and TiN NT. How do you drive the numerical values of the impedance (the impedance difference mentioned in page 14, line 243-245)? I recommend providing an equivalent circuit for the EIS data.

7. The full cell system is not actually an all solid-state battery because the cathode was impregnated with ionic liquid. To clearly indicate this, the "solid-state batteries" in the title should be revised to "hybrid solid-state battery".

Response to Reviewer #1

[Remarks] In this work, the authors employ the titanium nitrate nanotube structure combined with an Ag-C interlayer to address the anisotropic stress arising from repetitive Li deposition layers during cycling. By incorporating an MIEC material into the anode interface, the author emphasizes the enhanced movement of Li within the nanotube arrays, mainly due to diffusional creep during charge/discharge cycles. Remarkably, the battery demonstrated excellent performance. The article is well-constructed, possesses a unique perspective and provides new insight and methodologies. However, certain expressions and explanations are not very clear. In my view, several critical questions must be addressed before proceeding further.

Response: We appreciate your specific and constructive comments aimed at improving our manuscript. In response to your suggestions, we have conducted additional experiments to substantiate our explanations and have made every effort to enhance clarity in our expressions. All the changes shown in the following point-by-point responses are added to our revised manuscript.

#1-1. The authors claim that the effect of deposition behavior of lithium metal and suppressing volume expansion occurs via the interfacial-diffusional Coble creep mechanism. How does the author prove that lithium metal behaves as Coble creep mechanism rather than Nabarro-Herring creep in MIEC? In addition, the rate of diffusion creep is greatly affected by temperature. The authors should include a discussion of diffusion creep when studying electrochemical performance at different temperatures (Figure 5).

Response: We thank the reviewer for valuable comments and suggestions. We proposed that the Li_{bcc} deposition/stripping in TiN NT (MIEC) is governed by interfacial-diffusional Coble creep rather than Nabarro-Herring creep due to the following major criteria:

- 1) It has been found that a critical size for the inner diameter of the porous structures is distinguishing factor that determines the dominant creep mechanism. Referring to the tensile experiment that was conducted using Sn_{hcp} nano ligaments (as Sn_{hcp} and Li_{bcc} possess a similar homologous temperature of $T/T_M \sim 0.6$ (Figure R3a), the MIEC tubule with critical size of 100-200 nm is necessary for Coble creep to occur, and ~200-500 nm for Nabarro-Herring creep to occur (*Chem*, 2020, 6, 2878–2892). The TiN NT used in our work has an average inner diameter of ~100 nm, which is well within the critical size range for Coble creep to occur (Figure R3b).

- 2) When a metal's characteristic dimension is sufficiently confined to the nanoscale, specifically within a length scale of ~ 100 nm, the "smaller is stronger" trend of dislocation plasticity wear off, and the metal falls into a "smaller is weaker" Coble creep regime (**Figure R3c**) (*Adv. Mater.*, 2023, 35, 2210835). This implies that the Li_{bcc} may behave like an "incompressible work fluid" during battery cycling and the nanosized diameter tube can facilitate the fluidization of the Li_{bcc} thereby reducing its plastic hardness $\ll 10$ MPa.
- 3) Coble creep dominates via interfacial diffusion along the MIEC/ Li_{bcc} incoherent interface. TiN NT satisfies this condition in two main ways:
 - (i) MIEC material: TiN, the major building block of TiN NT has a high electrical conductivity as well as lithiophilic property (high Li adsorption energy and low diffusion energy barrier), owing to a facile charge transfer between Li and N atom (where Li acts as an electron donor and N, which resides in the interstitial space of Ti crystal acts as an electron acceptor). Furthermore, TiN NT is thermodynamically stable against Li_{bcc} (**Figure R3d**). Thus, these properties qualify TiN NT as an MIEC material (*Adv. Funct. Mater.*, 2019, 29, 1903229).
 - (ii) MIEC/ Li_{bcc} incoherent interface: the incoherent interface between MIEC and Li_{bcc} is another factor that contributes to the Li deposition/stripping via Coble creep. Since, TiN has a characteristic face-centered-cubic (fcc) structure (N atoms fit into octahedral sites in the Ti lattice), it forms an incoherent interface with Li_{bcc} that allows for a facile diffusion flux along the MIEC/ Li_{bcc} interface or the MIEC surface over the MIEC wall itself (**Figure R3e**) (*Hydrogen Energy*, 2020, 45, 28294–28302).

Consequently, these features corroborate that the Li_{bcc} deposition/stripping in TiN NT (MIEC) is governed by interfacial-diffusional Coble creep rather than Nabarro-Herring creep.

Additionally, the influence of temperature on diffusion creep is a crucial factor that needs to be accounted for when discussing the electrochemical performance of the full-cell (**Fig. 5**). In particular, the temperature is a critical parameter that affects the diffusion creep rate of Li metal inside the nanosized tube (or pore) confinement. Since Li metal at room temperature has a

homologous temperature of $T/T_M = 0.66$, it exhibits appreciable diffusional creep and be considered as viscous in the low-stress limit. As a result, the Li_{ibcc} may behave as an “incompressible working fluid” that flows inside the MIEC tubules, driven by the overpotential and the mechanical pressure gradient (*Nature*, 2020, 578, 251–255). If interfacial diffusional Coble creep is operative, the creep strain rate ($\dot{\epsilon}(T, \sigma)$) of Li_{ibcc} follows $\dot{\epsilon}(T, \sigma) \propto \sigma$, the viscosity η would depend on T and grain size (but not on σ), and Li would behave like a Newtonian fluid.

As the increase in the absolute temperature directly impacts the homologous temperature and the lowering of η , it would facilitate the Li_{ibcc} flow within the nanotube. Moreover, higher temperature leads to greater kinetic energy in atoms, enabling more frequent and longer jumps, thereby increasing the diffusion rate. This effect is manifested in the electrochemical performance test result of TiN NT incorporated AFSSB full-cell at 60 °C in Fig. 5d wherein the full-cell exhibited a more stable cyclability and coulombic efficiency compared to that of the full-cell cycled at 25 °C. There is also a concurrent effect of increase in the ionic conductivity of LLZTO solid electrolyte under high temperature condition that contributed towards the enhanced performance.

Figure R3. a) Schematic of the deformation mechanism map. b) Schematic of the design details with respect to the MIEC materials, architectures, sizes, and the lithiophilic coatings (*Chem*, 2020, 6, 2878–2892). c) Semiquantitative illustration of the deformation mechanism map for

metal (i.e., Sn). The gray curve denotes displacive deformation and surface diffusional deformation is governed by Coble creep (*Adv. Mater.*, 2023, 35, 2210835). d) Phase diagram of N₂–Li–Ti system, TiN (marked as red) lies in the thermodynamically stable phase against Li. e) Schematic for the mechanism of Coble creep that transport via interfacial diffusion along the MIEC/Li_{bcc} incoherent interface (Li et al., 2022, U.S. Patent No. 0208410 A1). (This Figure is for reviewers only).

Page 15

Original manuscript

The schematic of the fabricated full cell is shown in Fig. 5a. To evaluate the effect of using TiN NTs on the rate performance of the full cell, the rate capability was examined with varying current densities ranging from 0.3 mA cm⁻² to 4.0 mA cm⁻² at temperatures of 60 °C and 25 °C, respectively (Figs. 5b and c). As shown in Figs. 5b and c, the TiN NT-incorporated full-cells demonstrated a stable discharge capacity under a high current density of 4.0 mA cm⁻² at both 60 °C and 25 °C. The corresponding charge/discharge voltage profile of the full-cells (Supplementary Fig. 14) also shows characteristic voltage curves of the NCM333 cathode, which were sustained without short-circuit at 4.0 mA cm⁻².

Revised manuscript

The schematic of the fabricated full cell is illustrated in Fig. 5a. To evaluate the rate performance of the full cell, a rate capability test was conducted with varying current densities, ranging from 0.3 mA cm⁻² to 4.0 mA cm⁻², under two distinct temperature conditions: 60 °C and 25 °C (Fig. 5b, c). The utilization of different temperature settings aimed to examine the impact of temperature on the diffusion creep rate of Li metal inside the nanotube confinement. As shown in the results, both full-cells delivered a stable discharge capacity even at a high current density of 4.0 mA cm⁻². However, the rate capability of the full-cell at 60 °C consistently outperformed that at 25 °C across all current density conditions (Supplementary Fig. 19). This improved performance at elevated temperature can be attributed to various factors, including the accelerated Li metal creep rate within the nanotube due to the enhanced kinetics and the concurrent effect of an increase in the ionic conductivity of LLZTO.

#1-2. In Figures 3a-c, a new experimental technique was employed to study the Li propagation through the Ag-C interlayer. However, the data presented seems insufficient to provide a full understanding of the Li propagation behavior. Could the authors present evidence showing a reduction in residual lithium metal after introducing MIEC using this method? Additionally, could the presence of the copper mesh obscure some OM details because of the lithium metal deposited on it? Furthermore, there is an inconsistency between the "Ni mesh" label in Fig 3a and the mention of "Cu mesh" in the text.

Response: We thank the reviewer for their valuable comments. Firstly, as the reviewer mentioned, preventing lithium deposition at the interface between Ag-C and the negative electrode is crucial in minimizing volume changes of AFSSB during cell operation. As proven by the TOF-SIMS and in-situ OM analysis, TiN NT demonstrates the ability to accommodate Li_{bcc} within its nanotubes during the charging process, operating reversibly with minimal volume changes compared to Cu or Li metal anode counterparts. However, acknowledging the need for a comprehensive understanding of Li propagation behavior, we concur that clear evidence is required to demonstrate a reduction in residual lithium metal at the interface between Ag-C and TiN NT post-charging. To address this, we conducted a cross-sectional FIB-SEM analysis of the charged cell, specifically aiming to detect any residual lithium metal at the interface (Figure R4a) and the EDS elemental mapping was performed to delineate each region of the layer. As shown in the results, the cross-section SEM-EDS images reveal distinct regions corresponding to TiN NT (Figure R4b, c) and Ag-C layers (Figure R4e, f), characterized by a homogeneous distribution of their component elements. Furthermore, the presence of O element in both FIB-milled TiN NT and Ag-C regions indicates the existence of Li (in oxide form due to air exposure) (Figure R4d), which are expected to manifest as Li_{bcc} deposition within TiN NT and AgLi alloy within the Ag-C, respectively (Figure R4g). This result accentuates that the amount of residual lithium metal at the interface is negligible and also highlights the effective Li_{bcc} accommodating capability of TiN NT.

Figure R4. a) Cross-sectional FIB-SEM image, and b-g) Corresponding EDS elemental mapping images of the TiN NT-incorporated AFSSB after charging. (This Figure is included as Supplementary Fig. 10 in the revised manuscript).

Secondly, the reviewer raised concerns about the potential of the Cu mesh to obscure certain OM details due to the deposition of Li metal on the Cu mesh. In response to this concern, the authors would like to clarify the role of the Cu mesh. Cu mesh was introduced to serve dual purposes, acting both as the current collector to guide electron flow to the external circuit and as a window to enable an in-situ analysis of Li metal propagation through the Ag-C interlayer. To scrutinize this deposition behavior more closely, the SEM image of the charged cell was

taken after the OM analysis (Figure R5a). As shown in the top SEM image (Figure R5b), the stream of Li metal protruding on the surface of the Ag-C interlayer can be detected adjacent to the Cu mesh. The cross-section SEM image at this border taken via focused ion beam (FIB) milling, reveals a layer of deposited Li metal between the Cu mesh and Ag-C (Figure R5c). This observation was further substantiated by EDS elemental mapping, showing an O K α rich region indicative of an oxidized Li layer due to air exposure (Figure R5d). Furthermore, employing a punched Cu foil, which has a smaller hole diameter than the Cu mesh resulted in the visibility of Li metal projection only near the edge of the punched hole (Figure R6). This underscores the necessity of a larger hole size for direct observation of Li propagation from the Ag-C, making the Cu mesh more suitable for in-situ OM analysis. Therefore, the Cu mesh's role as a window and current collector proved essential in providing a comprehensive understanding of Li metal propagation. Lastly, we apologize for mislabeling the "Cu mesh" as "Ni mesh" in Fig. 3a of our manuscript. In response to the reviewer's valuable comment, we have added further details and corrections in **Supplementary information** and added explanations in the manuscript, as follows:

Figure R5. a) Digital image and schematic of the cell configuration used for in-situ OM analysis. b) Top SEM images of the charged cell after the in-situ OM experiment. c) Cross-section FIB-SEM images of the edge side of Cu mesh revealing layers of Li metal and Ag-C. d)

Corresponding EDS elemental mapping image of the marked region. (This Figure is included as Supplementary Fig. 13 in the revised manuscript).

Figure R6. a) Top SEM image showing the negative electrode of the charged full-cell, utilizing a punched Cu foil as a current collector, with each hole measuring $\sim 350 \mu\text{m}$ in diameter. b) Magnified SEM image of the highlighted region, revealing a Li metal projection at the edge of the punched hole. (This Figure is for reviewers only)

Page 8

Original manuscript

In contrast, for the case of 100% SOC, a strong Li^+ signal was detected throughout the sputtering time, indicating that Li metal was successfully deposited inside the TiN NT capillaries over the course of the charging process. These results suggest that the TiN NT anode can effectively accommodate Li metal inside its porous structure, mitigating volume expansion in the interlayer between the Ag-C and the current collector. This enables strain-free operation, as minimized volume change can prevent associated mechanical stress and deformation applied to the cell during the charging/discharging process.

Revised manuscript

In contrast, for the case of 100% SOC, a strong Li^+ signal was detected throughout the sputtering time, indicating that Li metal was successfully deposited inside the TiN NT capillaries over the course of the charging process. These results suggest that the TiN NT anode can effectively accommodate Li metal inside its porous structure, mitigating volume expansion in the interlayer between the Ag-C and the current collector. This enables strain-free operation, as minimized volume change can prevent associated mechanical stress and deformation applied to the cell during the charging/discharging process. **Furthermore, the**

cross-sectional focused ion beam (FIB)-SEM analysis of the charged cell was performed to detect any residual lithium metal at the interface (Supplementary Fig. 10a). The EDS elemental mapping images reveal distinct regions corresponding to TiN NT (Supplementary Fig. 10b, c) and Ag-C layers (Supplementary Fig. 10e, f), characterized by a homogeneous distribution of their component elements. Evidently, the presence of O element in both FIB-milled TiN NT and Ag-C regions indicates the existence of Li (in oxide form due to air exposure), which are expected to manifest as Li_{bcc} deposition within TiN NT and AgLi alloy within the Ag-C, respectively (Supplementary Fig. 10d, g). This result accentuates that the amount of residual lithium metal at the interface is negligible and also highlights the effective Li_{bcc} accommodating capability of TiN NT.

Page 10

Original manuscript

To conduct operando OM analysis, we utilized a specially designed cell consisting of a Cu mesh (opening size: $300\ \mu\text{m} \times 300\ \mu\text{m}$) current collector to monitor the Li evolution in the Ag-C interlayer during the charging/discharging process (Fig. 3a and Supplementary Fig. 8). To improve the cell's cyclability, we employed a new pre-lithiation technique that involves attaching a halo Li foil onto the Ag-C coating layer via CIP under 250 MPa. Due to the facile alloying property of Ag to form Ag-Li, partial physical contact between Ag-C and Li was sufficient for pre-lithiation to occur^{36,37}. As evidence, the cell with the pre-lithiated Ag-C (Pre_Li_Ag-C) interlayer exhibited superior Li plating and stripping performance compared to that with the pure Ag-C interlayer (Supplementary Fig. 9), along with enhanced specific capacity and reversibility in the first cycle. Using this setup, we observed the Li evolution from the Ag-C interlayer under an applied galvanostatic current of $1.0\ \text{mA cm}^{-2}$ at $25\ ^\circ\text{C}$. A real-time observation of the Ag-C layer during the charging process (Fig. 3b) shows that island-type Li precipitates can be found on the Ag-C layer after 60 min, and they grow in size and number after 120 min of deposition (Fig. 3c). As more clearly shown in Supplementary Video 1, Li begins to protrude from the surface of the Ag-C layer in the form of interspersed islands and continues to grow over time during charging. It then develops into filamentary Li as evidenced by the out-of-focus features in the OM image caused by the elevation of height above the Ag-C surface. During the subsequent discharging process, the

Li deposits on the Ag-C layer gradually recede back into the Ag-C layer, but some remaining Li islands can be observed at the end, revealing incomplete stripping of Li at the interface.

Revised manuscript

To conduct *operando* OM analysis, we utilized a specially designed cell consisting of a Cu mesh (opening size: 300 $\mu\text{m} \times 300 \mu\text{m}$) current collector to monitor the Li evolution in the Ag-C interlayer during the charging/discharging process (Fig. 3a and Supplementary Fig. 12a). To improve the cell's cyclability, we employed a new pre-lithiation technique that involves attaching a halo Li foil onto the Ag-C coating layer via CIP under 250 MPa. Due to the facile alloying property of Ag to form Ag-Li, partial physical contact between Ag-C and Li was sufficient for pre-lithiation to occur^{36,37}. As evidence, the cell with the pre-lithiated Ag-C (Pre_Li_Ag-C) interlayer exhibited superior Li plating and stripping performance compared to that with the pure Ag-C interlayer (Supplementary Fig. 13), along with enhanced specific capacity and reversibility in the first cycle. Using this setup, we observed the Li evolution from the Ag-C interlayer under an applied galvanostatic current of 1.0 mA cm^{-2} at 25 °C. A real-time observation of the Ag-C layer during the charging process (Fig. 3b) shows that island-type Li precipitates can be found on the Ag-C layer after 60 min, and they grow in size and number after 120 min of deposition (Fig. 3c). As more clearly shown in Supplementary Video 1, Li begins to protrude from the surface of the Ag-C layer in the form of interspersed islands and continues to grow over time during charging. It then develops into filamentary Li as evidenced by the out-of-focus features in the OM image caused by the elevation of height above the Ag-C surface. Moreover, a closer examination of the Ag-C surface via SEM analysis reveals a stream of Li metal protruding adjacent to the Cu mesh (Supplementary Fig. 12b). Cross-sectional FIB-SEM and EDS image at this border disclosed a layer of deposited Li metal between the Cu mesh and Ag-C (Supplementary Fig. 12c, d). During the subsequent discharging process, the Li deposits on the Ag-C layer gradually recede back into the Ag-C layer, but some remaining Li islands can be observed at the end, revealing incomplete stripping of Li at the interface.

#1-3. Figure 2 presents the lithium metal deposition process. Would the author consider adding pertinent ion fragmentation data for TiN to further substantiate this experiment?

Response: Thank you for your insightful suggestion. In our study, to confirm the localization of Li metal within TiN NT based on the state of charge, we extracted the TiN NT anode at 20%

and 100% SOC. Subsequently, we conducted TOF-SIMS analysis by etching it with an O_2^+ sputter beam from the top side of TiN NT to investigate the depth profile of various secondary ions.

As shown in Figure R7, the secondary ion depth profile of TiN NT after charging exhibited Ti^+ and NH_3^+ secondary ions corresponding to the TiN and Li^+ secondary ions corresponding to deposited Li, respectively. Notably, with a sputtering time exceeding $\sim 22,500$ seconds, the signals of NH_3^+ and Li^+ ions disappeared, indicating that the etched TiN NT reached the Ti foil (current collector), and only the Ti^+ secondary ion signal remained. Consequently, we successfully determined the location of Li metal within TiN NT at each SOC. Figure R7, and its related discussions are reflected in the revised manuscript (page 6) and Supplementary information (Supplementary Fig. 7).

Figure R7. Normalized TOF-SIMS depth profile of various secondary ions for TiN NTs at a) 20% SOC and b) 100% SOC. (This Figure is included as Supplementary Fig. 7 in the revised manuscript).

Original manuscript

Fig. 2a shows the normalized depth profile of the Li^+ secondary ion signal, representing the relative intensity of the deposited Li metal along the depth of the TiN NTs for each charged sample. As the sputtering time increased, the intensity of the Li^+ secondary ion signal tended to decrease, suggesting that the amount of Li metal within the TiN NT was most abundant near the LLZTO/TiN NT interface and progressively decreased in the direction towards the current collector during the charging process.

Revised manuscript

Fig. 2a shows the normalized depth profile of the Li^+ secondary ion signal, representing the relative intensity of the deposited Li metal along the depth of the TiN NTs for each charged sample. Beyond a sputtering time of approximately 22,500 seconds, the signals of NH_3^+ and Li^+ ions disappeared, leaving only the Ti^+ secondary ion signal, indicating that the etched TiN NT reached the Ti foil (current collector) (Supplementary Fig. 7a, b). It is worth noting that as the sputtering time increased, the intensity of the Li^+ secondary ion signal tended to decrease. This observation suggests that the concentration of Li metal within the TiN NT was most abundant near the LLZTO/TiN NT interface and progressively decreased in the direction towards the current collector during the charging process.

#1-4. The authors should explain more specifically how Fig 4f combine the symmetric cells with “volume change buffering capability”. The current rationale is not persuasive.

Response: We thank the reviewer for their kind and insightful comments. We acknowledge that our current rationale, which combines the volume change buffering capability with the symmetric cell performance, is not sufficient. To address this concern, we conducted additional experiment using two pouch-type symmetric cells, each containing TiN NT and Cu with the pre-lithiated Ag-C (Pre_Li_Ag-C) interlayer (denoted simply as TiN NT and Cu symmetric cells, respectively). Both cells were encased in vacuum-sealed laminated Al pouches. We then compared the thickness change of these cells during the initial charging process (0.2 C under 25 °C) using a laser displacement measurement setup (Figure R8a). As shown in the results (Figure R8b), the thickness change of the TiN NT symmetric cell was less pronounced than that of Cu counterpart. By the end of the charging process, the thickness of Cu symmetric cell

reached 20 μm , whereas the TiN NT symmetric cell only reached the thickness of 12 μm , which is nearly half of the Cu symmetric cell. These results show that the volume buffering capability of TiN NT contributes towards the prolonged cycling of TiN NT symmetric cell. Figure R8 and its related discussions are reflected in the revised manuscript (page 14) and Supplementary information (Supplementary Fig. 17).

Figure R8. a) Charge voltage profile of Cu and TiN NT incorporated symmetric cells. b) Corresponding thickness changes of two symmetric cells as a function of charging time. (This Figure is included as Supplementary Fig. 17 in the revised manuscript).

Page 14

Original manuscript

To further confirm the volume change buffering capability of the TiN NTs, we fabricated two different symmetric cells using Pre_Li_Ag-C/Cu and Pre_Li_Ag-C/TiN NTs and carried out Li plating/stripping tests under varying current densities at 25 °C (Fig. 4f). The figures show that the critical current density was significantly improved by employing the Pre_Li_Ag-C/TiN NT, as the Pre_Li_Ag-C/Cu symmetric cell showed a continuous increase in polarization, which ultimately led to a short circuit at a current density of 1.6 mA cm⁻².

Revised manuscript

To further investigate the volume change buffering capability and electrochemical performance of TiN NT during harsh lithium deposition/stripping conditions, we fabricated two symmetric cells, each containing TiN NT and Cu attached to opposite sides of the Pre_Li_Ag-C interlayer double-affixed LLZTO (hereafter referred to as TiN NT and Cu symmetric cells). Subsequently, we measured their thickness change during the initial

charging process (0.2 C) using a laser displacement measurement setup (Supplementary Fig. 17a). The results show that by the end of the charging process, the thickness of Cu symmetric cell reached 20 μm , whereas the TiN NT symmetric cell only reached 12 μm , which is nearly half of the Cu symmetric cell (Supplementary Fig. 17b). This validates the efficient volume buffering capability of TiN NT. Additionally, Li plating/stripping tests were carried out under varying current densities at 25 °C. The result in Fig. 4f demonstrates that the Cu symmetric cell undergoes a continuous increase in polarization at a current density of 1.6 mA cm^{-2} and ultimately succumbs to a short circuit in less than 40 h. In contrast, the TiN NT symmetric cell delivers stable stripping/deposition cycles without noticeable overpotential, indicating an improvement in the critical current density.

#1-5. The authors' impedance analysis seems oversimplified. Given the presence of multiple phases in the system, there might be numerous interfaces and associated electrochemical process responses. Introducing temperature as a variable further complicates the analysis. Thus, relying solely on the comparison of "the size of the semicircle" appears inadequate. It would be beneficial if the fitting circuits for Fig 5e and s113 were provided. Highlighting the relevant electrochemical processes and subsequently drawing a detailed comparison would make the discussion more comprehensive.

Response: We thank the reviewer for their valuable suggestions. Following the suggestions of the reviewer, we calculated the ohmic resistance (R_b), interfacial resistance (R_{ct}), and Warburg resistance (W_R) by complex nonlinear least-squares fitting of the experimental impedance spectra in Fig. 5e and Supplementary Fig. 13 based on a given equivalent circuit. We have added images of the equivalent circuit and listed the resultant fitting values to Supplementary Table 2 and 3 to facilitate a better comparison of R_b , R_{ct} , and W_R .

Table R1. Fitting parameters determined from CNLS fitting of the measured impedance spectra in Supplementary Fig. 18. (This Table is included as Supplementary Table 2 in the revised manuscript).

[Symmetric cell (Pristine)]

Fitting parameters	Anode			
	Cu		TiN NT	
	Fit values	Error (%)	Fit values	Error (%)
R_b [$\Omega \text{ cm}^2$]	18.67	0.36	18.85	0.42
R_1 [$\Omega \text{ cm}^2$]	5.76	2.52	3.96	2.81
CPE_{1-T}	6.12×10^{-6}	19.17	3.17×10^{-5}	20.44
CPE_{1-n}	0.86	2.38	0.77	3.00
chi-square	5.1×10^{-4}		9.68×10^{-4}	

[Symmetric cell (After cycle)]

Fitting parameters	Anode			
	Cu		TiN NT	
	Fit values	Error (%)	Fit values	Error (%)
R_b [$\Omega \text{ cm}^2$]	18.08	0.57	18.44	0.48
R_1 [$\Omega \text{ cm}^2$]	9.03	1.65	5.56	2.27
CPE_{1-T}	1.36×10^{-5}	14.26	2.92×10^{-5}	17.94
CPE_{1-n}	0.75	2.02	0.74	2.70
chi-square	1.78×10^{-3}		1.45×10^{-3}	

Table R2. Fitting parameters determined from CNLS fitting of the measured impedance spectra in Fig. 5e. (This Table is included as Supplementary Table 3 in the revised manuscript).

[60 °C]

Fitting parameters	Anode			
	Cu		TiN NT	
	Fit values	Error (%)	Fit values	Error (%)
R_b [Ω cm ²]	7.34	0.11	5.0	0.15
R_l [Ω cm ²]	14.10	0.93	12.75	0.93
R_2 [Ω cm ²]	2.54	2.76	2.41	2.35
$R_{ct} = R_l + R_2$ [Ω cm ²]	16.64		15.16	
CPE ₁ -T	1.85×10^{-5}	2.54	1.10×10^{-5}	2.67
CPE ₁ -n	0.76	0.33	0.87	0.31
W _R [Ω cm ²]	14.44	10	63.87	11.22
W-T	0.076	10.19	0.25	13.82
W-n	0.40	0.79	0.37	1.38
CPE ₂ -T	4.8×10^{-6}	15.85	2.44×10^{-6}	16.07
CPE ₂ -n	0.89	4.05	0.95	3.44
chi-square	1.35×10^{-4}		2.19×10^{-4}	

[25 °C]

Fitting parameters	Anode			
	Cu		TiN NT	
	Fit values	Error (%)	Fit values	Error (%)
R_b [$\Omega \text{ cm}^2$]	31.06	0.16	22.01	0.2
R_1 [$\Omega \text{ cm}^2$]	25.98	4.39	12.34	3.98
R_2 [$\Omega \text{ cm}^2$]	81.39	1.21	49.74	2.79
$R_{ct} = R_1 + R_2$ [$\Omega \text{ cm}^2$]	107.37		62.08	
CPE ₁ -T	9.3×10^{-6}	8.95	3.17×10^{-6}	12.6
CPE ₁ -n	0.73	1.11	0.87	1.48
W_R [$\Omega \text{ cm}^2$]	61.97	3.85	63.87	3.95
W-T	0.29	5.56	0.25	6.3
W-n	0.40	0.99	0.37	1.92
CPE ₂ -T	5.75×10^{-6}	2.37	2.44×10^{-6}	5.95
CPE ₂ -n	0.86	0.07	0.95	1.18
chi-square	3.38×10^{-4}		2.60×10^{-4}	

Figure R9. Nyquist plots of AC-impedance spectra obtained from the symmetric cells with different anodes (Cu vs. TiN NT) before and after cycle (inset: equivalent circuit). a) Before cycle and b) After cycle. (This Figure is included as Supplementary Fig. 18 in the revised manuscript).

Figure R10. Nyquist plots of AC-impedance spectra obtained from the AFSSB full-cells with different anodes (Cu vs. TiN NT) at different temperature conditions (inset: equivalent circuit). a) 25 °C and b) 60 °C. (This Figure is included as Fig. 5e and Supplementary Fig. 20 in the revised manuscript).

Page 15

Original manuscript

The electrochemical impedance spectroscopy (EIS) result (Supplementary Fig. 13) also revealed that the total resistance could be reduced from 3.3 $\Omega \text{ cm}^2$ to 2.6 $\Omega \text{ cm}^2$ for the pristine cell and from 8.4 $\Omega \text{ cm}^2$ to 5.0 $\Omega \text{ cm}^2$ for the cycled cell by employing Pre_Li_Ag-C/TiN NT.

Revised manuscript

Moreover, the electrochemical impedance spectroscopy (EIS) measurement of the same symmetric cells before and after cycle was conducted to evaluate their change in the cell impedance (Supplementary Fig. 18a, b). The Nyquist plot and corresponding fitting result (Supplementary Table 2) revealed that the TiN NT symmetric cell exhibits much lower interfacial resistance compared to its Cu counterpart before cycle (3.96 vs. 5.76 $\Omega \text{ cm}^2$) and the trend persists after the cycle (5.56 vs. 9.03 $\Omega \text{ cm}^2$), signifying the efficacy of the TiN NT in maintaining the interfacial contact during cell operation.

Original manuscript

~ Additionally, we conducted EIS to analyze the effect of the TiN NTs on the impedance of the cell and compared it with that of the Cu current collector under two temperature conditions (60 °C and 25 °C). Notably, the impedance difference between the TiN NT and Cu installed full cells was negligible at 60 °C (9.2 Ω cm² vs. 11.2 Ω cm²), while there was a significant difference in the impedance at 25 °C (55.3 Ω cm² vs. 108.7 Ω cm²) (Fig. 5e). This large disparity can be ascribed to the increase in interfacial resistance due to the reduced ionic conductivity of LLZTO SE at room temperature. Thus, this signifies that the current collector interface is a dominating factor for cell operation at low temperature (25 °C).

Revised manuscript

~ Additionally, we conducted EIS analysis on TiN NT and Cu installed full-cells at both 25 and 60 °C to comparatively examine their impact on cell impedance under difference temperature conditions. Fig. 5e and Supplementary Fig. 20 shows the Nyquist plots of the impedance spectra consisting of two depressed arcs, which overlap in the high frequency range, Warburg impedance, and a capacitive line (blocking region) in the low-frequency range. The equivalent circuit was implemented to quantitatively analyze the experimental impedance spectra, and the complex nonlinear least-squares (CNLS) fitting method was used to determine the bulk ionic resistance (R_b) and interfacial resistance ($R_{ct} = R_1 + R_2$) (the values are listed in Supplementary Table 3). Notably, a more pronounced difference in R_b and R_{ct} values between the TiN NT and Cu full-cells occurs at 25 °C (than at 60 °C) due to the overall increase in bulk and interfacial resistance associated with the lower ionic conductivity of the LLZTO SE at room temperature. Amidst of this limited ionic conduction, the impedance contribution at the Ag-C and the current collector interface becomes dominant. This is corroborated by the significantly lower R_b and R_{ct} values (22.0 and 62.1 Ω cm²) of the TiN NT full-cell compared to its Cu counterpart (31.1 and 107.4 Ω cm²), which can be ascribed to superior charge-transfer kinetics and interfacial stability at the Ag-C|TiN NT interface.

Response to Reviewer #2

[Remarks] This manuscript describes a 3-D mixed conducting host structure in conjunction with a Ag-C interlayer to manage Li plating and stripping in garnet-based solid-state batteries. The results are a significant advance over the state-of-the-art and are well-presented. Therefore, I am happy to recommend publication after the following comments are addressed.

Response: We thank you for your helpful and detailed comments to further improve our manuscript. We have carefully addressed your valuable comments in the following point-by-point responses, and the corresponding changes have been incorporated into our revised manuscript.

#2-1. Operando should be italicized when used.

Response: Thank you for your comment. We carefully reviewed the manuscript and italicized the word ‘Operando’.

#2-2. What is the internal pressure of the cells used? How does it change during cycling? Could these changes impact performance, and how would they be different depending on form factor of the cells? Reducing stack pressure requirements is key to SSB development, so it should be addressed here. Does this approach to minimizing thickness change also change pressure requirements?

Response: We express our gratitude to the reviewer for their kind and insightful comments. In the electrochemical characterization experiments, a conventional CR2032 coin-type full-cell was used with a configuration as illustrated in Figure R11a. The cell components were assembled via clamping, resulting in a stack pressure of approximately 0.6 MPa. For volume change measurement tests, a pouch-type cell with a similar cell configuration was utilized. However, in this setup, the internal pressure was generated solely by the vacuum inside the pouch (~1 bar), with no external stacking pressure applied (Figure R11b).

The primary cause of internal strain in SSBs is the volume change/deformation of their cell components, particularly the cathode and anode. To assess this strain generation in SSBs, we opted for non-contact detection methods, specifically employing a laser displacement sensor to measure the thickness change of the full-cell during charging/discharging. This approach provides more accurate information about the intrinsic volumetric expansion/contraction of the materials compared to contact-type measurements, such as those using a load cell. Hence, the

pouch-type cell configuration was chosen to confine the cell under near-zero stack pressure condition.

Evidently, our experimental setup with the laser displacement sensor revealed the superior volume/strain buffering capability of TiN NT, showcasing effective Li_{bcc} accommodation within the nanotube arrays. Notably, the analysis demonstrated that the cell could be cycled without incurring internal short circuits or capacity fading, even under near-zero stack pressure condition. This characteristic distinguishes it from other counterparts which experiences either short circuits (as observed in Ag-C/Li-metal) or constant capacity fading (as observed in Pre_Li_Ag-C/Cu). The latter issues may arise due to imperfect contact and consequent interfacial contact loss during cycling. Therefore, minimizing the thickness change of SSBs is crucial for enabling the cell to operate effectively under low stack pressure conditions.

Figure R11. a) Schematic of the coin-cell configuration used for the electrochemical characterizations. b) Schematic of the pouch-cell configuration used for the thickness change analysis during the charge/discharge process. (This Figure is included as and Supplementary Fig. 15 in the revised manuscript).

Page 12

Original manuscript

Additionally, the cell was vacuum sealed to prevent any external interference and was paired with a potentiostat to control the charging/discharging cycles (Supplementary Fig. 11). Using this setup, we compared three pouch type full-cells with different interlayer/anode configurations: Ag-C/Li, Pre_Li_Ag-C/Cu, and Pre_Li_Ag-C/TiN NT. We first examined the change in thickness of each full cell when they were charged under the same current density of 0.64 mA cm^{-2} (Fig. 4b). The results in Fig. 4c show that by the end of the charging time, the pouch cell with the Pre_Li_Ag-C/TiN NT exhibited approximately 85% suppression in Li deposition-induced volume expansion compared to the other two cells.

Additionally, the change in thickness of the pouch cells from their initial state was measured as a function of cycle number (Fig. 4d). The pouch cell with Ag-C/Li increased significantly in thickness during cycling, leading to a short circuit after the 7th cycle, which might be attributable to the mechanical failure of the LLZTO SE caused by local disintegration at the interface. In the case of the pouch cell with the Pre_Li_Ag-C/Cu, the thickness increased rapidly in the first cycle, but reached a plateau afterwards with a consistent increment from the initial state. In contrast, the pouch cell with the Pre_Li_Ag-C/TiN NT decreased in thickness after the first cycle, presumably due to the intratubular diffusion process, and then it maintained its thickness with only a minute variation throughout the measuring cycles. Notably, the Pre_Li_Ag-C/TiN NT-incorporated pouch cell exhibited an average change in thickness of 3.5 μm from its pristine state, a change that is significantly lower than that of the Ag-C/Li (42.1 μm) and Pre_Li_Ag-C/Cu (15.1 μm) installed pouch cells (Supplementary Table 1). Additionally, we compared the capacity retentions of the pouch cells from the above test, and the result in Fig. 4e shows that Pre_Li_Ag-C/TiN NT exhibited an outstanding capacity compared to the other two cells. Interestingly, the capacity of the Pre_Li_Ag-C/TiN NT mounted pouch cell increased consistently until the 7th cycle, presumably due to the emergence of the previously deposited Li in the nanotubes in the subsequent discharging step. It has been reported that the creeping of Li into the nanotubes can hermetically seal the structure, resulting in an internal pressure difference that might lead to heterogeneity in Li filling.

Revised manuscript

Additionally, the cell was vacuum sealed to prevent any external interference and paired with a potentiostat for cycling control. The internal pressure was generated solely by the vacuum inside the pouch (~1 bar), with no external stacking pressure applied (Supplementary Fig. 15). Using this setup, we compared three pouch type full-cells with different interlayer/anode configurations: Ag-C/Li, Pre_Li_Ag-C/Cu, and Pre_Li_Ag-C/TiN NT. We first examined the change in thickness of each full cell under the same charging current density of 0.64 mA cm^{-2} (Fig. 4b). The results in Fig. 4c demonstrate that the pouch cell with the Pre_Li_Ag-C/TiN NT exhibited approximately 85% suppression in Li deposition-induced volume expansion compared to the other two cells by the end of the charging time. Moreover, the change in thickness of the pouch cells from their initial state was measured as a function of

cycle number (Fig. 4d). The pouch cell with Ag-C/Li significantly increased in thickness during cycling, leading to a short circuit after the 7th cycle, which might be attributable to interfacial disintegration originating from the volume change during cycling. The pouch cell with the Pre_Li_Ag-C/Cu experienced rapid thickness increase in the first cycle but reached a plateau afterwards with a consistent increment from the initial state. In contrast, the pouch cell with the Pre_Li_Ag-C/TiN NT decreased in thickness after the first cycle, presumably due to the intratubular diffusion process, and then maintained its thickness with only minute variation throughout the measuring cycles. Notably, the Pre_Li_Ag-C/TiN NT-incorporated pouch cell exhibited an average change in thickness of 3.5 μm from its pristine state, significantly lower than that of the Ag-C/Li (42.1 μm) and Pre_Li_Ag-C/Cu (15.1 μm) installed pouch cells (Supplementary Table 1). Additionally, when measuring the change in thickness during the lithiation/delithiation process (Supplementary Fig. 16), the Pre_Li_Ag-C/TiN NT-installed pouch cell displayed the lowest average change in thickness (4.3 μm) compared to Ag-C/Li (16.3 μm) and Pre_Li_Ag-C/Cu (11.6 μm) (Supplementary Table 1). Furthermore, the comparison of capacity retentions among the pouch cells from the above test (Fig. 4e) highlights the outstanding performance of Pre_Li_Ag-C/TiN NT compared to the other two cells. Interestingly, the capacity of the Pre_Li_Ag-C/TiN NT mounted pouch cell increased consistently until the 7th cycle, presumably due to the emergence of the previously deposited Li in the nanotubes in the subsequent discharging step. It has been reported that the creeping of Li into the nanotubes can hermetically seal the structure, resulting in an internal pressure difference that might lead to heterogeneity in Li filling. Consequently, these analyses unequivocally demonstrate the superior volume/strain buffering capability of TiN NT, underscoring its effective Li_{bcc} accommodation within the nanotube arrays. It is also worth noting that the TiN NT incorporated full-cell could be cycled without incurring internal short circuits or capacity fading, even under near-zero stack pressure conditions. Therefore, minimizing the thickness change is a crucial factor for enabling the cell to operate effectively under low stack pressure conditions.

#2-3. How much excess Li remains in the Li halo used for pre-lithiation? Is this Li halo active during cycling? If used in practice, how much would this excess Li impact energy density (volumetric and gravimetric?)

Response: We appreciate your constructive comment. In our system, the pre-lithiation of the Ag-C interlayer is achieved through a cold isotactic pressing (CIP) process under 250 MPa

with the halo Li. Since the halo Li exhibits strong adhesion to the Ag-C interlayer during the pre-lithiation process, it becomes challenging to separate the remaining halo Li and measure its mass. Therefore, we conducted titration gas chromatography (TGC) analysis to confirm the amount of excess Li metal remaining after the pre-lithiation process (*Nature*, 2019, 572, 511, *Adv. Mater.*, 2023, 35, 2306826). The quantity of excess Li metal can be calculated by measuring the generated H₂ based on the chemical reactions below:

Firstly, a calibration curve was established by measuring the H₂ area generated from the reaction between a known weight of Li metal and H₂O. The amount of remaining Li metal in each sample was then calculated by measuring the H₂ area generated after the reaction with H₂O (Figure R12 and R13). There was no H₂ generation observed in LLZTO/Ag-C (before pre-lithiation), and in the case of the halo Li, the quantified amount of Li metal was 0.638 mg, closely matching the balance measurement (0.639 mg).

Additionally, the amount of remaining Li metal in the halo Li used in LLZTO/Pre__Li_Ag-C was calculated as 0.405 mg, indicating that a portion of the Li metal in the halo Li was consumed during the pre-lithiation of the Ag-C interlayer. Finally, after the charging/discharging cycle, the amount of Li metal in the halo Li of LLZTO/Pre_Li_Ag-C remains at 0.407 mg, consistent with the amount of excess Li metal before the cycle. These results indicate that the excess Li remaining after pre-lithiation does not participate in the charging/discharging cycle and does not affect the overall performance and energy density of the full cell.

Figure R12. Calibration curve of the detected H₂ area as a function of Li metal amount. (This Figure is for reviewers only).

Figure R13. Generated H₂ peaks after reactions with H₂O for (a) LLZTO/Ag-C, (b) pristine halo Li, (c) LLZTO/Pre_Li_Ag-C (before cycle), and (d) LLZTO/Pre_Li_Ag-C (after cycle). (This Figure is for reviewers only).

#2-4. What are the sources of degradation during cycling in these full cells? Why is impedance increasing over time?

Response: We appreciate the reviewer for their valuable comments. As pointed out by the reviewer, the long-term full-cell performance result in Fig. 5d, f is not completely stable over the 600 cycles. There could be multiple reasons that are responsible for the degradation of cell over repeated cycles. Some of the possible sources of degradation during full-cell cycling are listed below:

- 1. Degradation at the cathode side:** The full-cell that was used in long-term cycling test consists of ionic liquid (IL) wetted NCM333 cathode which is affixed to the LLZTO solid electrolyte. Over repeated cycling the NMC cathode materials are known to suffer from capacity fading, which has been correlated to increased cell polarization, phase transition at the surface (layered to rock salt), transition metal (TM) dissolution, etc. These are mostly caused by the side reaction at the cathode-electrolyte interface,

oxidation of electrolyte, leaching due to acidic species, etc. (*J. Electrochem. Soc.*, 168, 060518–060518). In addition, the IL that was used to wet NMC cathode was only employed in the initial cell assembly, and it was not refilled during the cycling process. Therefore, the accumulation of side reaction by-products and decomposition of IL over repeated cycles can potentially facilitate the interfacial impedance at the cathode-electrolyte interface and ultimately lead to the full-cell capacity fading.

- 2. Degradation at the LLZTO solid electrolyte:** The LLZTO solid electrolyte within the full-cell may undergo degradation over extended cycling periods, potentially influencing cell performance and cycle stability. One degradation process involves interfacial instability at the IL-wetted NCM-LLZTO interface. The decomposition of the IL over prolonged cycling can result in increased impedance and ionic contact loss at the cathode-electrolyte interface, leading to reduced overall cell performance. Additionally, microstructural changes such as grain boundary evolution or the development of defects within the LLZTO can impact its mechanical integrity and, subsequently, its electrochemical performance over time (*ACS Energy Lett.*, 2023, 8, 9–20). These changes could contribute to the observed degradation in the full-cell cycling performance.
- 3. Degradation at the anode side:** It is conjectured that there could be some possible sources of degradation at the anode side that are responsible for the full-cell capacity fading. One of them is the displacement of Ag particles within the Ag-C interlayer after prolonged cycling process. Numerous studies have demonstrated the effectiveness of the Ag-C interlayer, not only in improving interfacial contact but also in preventing direct contact between the solid electrolyte and the Li metal anode, thereby suppressing dendrite penetration during cell operation. Therefore, the use of an Ag-C interlayer is undeniably a crucial component in maintaining cycle stability in SSBs. However, it has also been observed that, over time, some Ag particles within the Ag-C interlayer migrate towards the Li metal owing to their thermodynamic preference for forming a Li-Ag alloy during cycling (*Nat. Commun.*, 2023, 14, 782). The redistribution of Ag within the Ag-C towards the interlayer interface via chemical diffusion along with the gradual loss of available alloying elements over time, although subtle, may contribute to the observed full-cell capacity fading during cycling.

Response to Reviewer #3

[Remarks] In this work, the authors proposed an MIEC-based TiN nanotube structure capable of accommodating lithium through diffusional creep, along with an Ag-C interlayer. In an anode-free system, repeated Li plating and stripping cause significant volume changes, posing critical problems when using oxide solid electrolytes. The authors addressed this strain issue by utilizing TiN nanotubes as a 3D host for strain-free anode architecture. Through this strategy, remarkable cyclability was achieved. The in-situ analyses are interesting, however, the author's claims on the role of TiN NT and Ag-C interlayer were not clearly supported in this manuscript. The improvement in performance could be merely from the use of ionic liquid in the cathode and excess-Li in so-called the pre-lithiation anode. I thereby recommend rejection of this manuscript. Questions and comments are provided below.

Response: We thank for your detailed and constructive comment. As highlighted by the reviewer, we were able to showcase an effective method for mitigating strain generated in a garnet-type solid electrolyte-based anode-free system during the repeated Li plating and stripping process via the TiN NT 3D host. The MIEC-based TiN NT has shown to be highly capable of accommodating Li metal within its nanotubes, facilitating a reversible lithiation/delithiation process without causing significant changes in cell thickness. However, it is important to note that this characteristic of TiN NT can only be realized when coupled with an Ag-C interlayer. For TiN NT to serve as an effective 3D host for Li_{bcc} , the interfacial contact at the LLZTO/TiN NT interface must be homogeneous and ideally defect-free. However, inherent surface irregularities in both LLZTO and TiN NT naturally prevent them from achieving complete contact, resulting in substantial interfacial resistance. Therefore, the utilization of the Ag-C interlayer is compulsory for the proper functioning of TiN NT in an LLZTO-based anode-free system. In addition, the pre-lithiation of Ag-C was carried out using a halo Li to enhance the initial coulombic efficiency and thereby improve the performance of anode-free NCM333 full-cell. Contrary to the concern raised by the reviewer, it was confirmed via titration gas chromatography (TGC) analysis that the excess Li remaining in the halo Li after the pre-lithiation process did not participate in the lithiation/delithiation process. Only the Li from the cathode and Pre_Li_Ag-C are engaged in the charge/discharge process. Furthermore, pre-lithiation of Ag-C promotes the adhesion at the LLZTO/TiN NT interface, resulting in more uniform interfacial contact after CIP. We acknowledge that these details were not adequately elaborated in the original manuscript. Therefore, additional information has

been included in the revised manuscript, and point-by-point responses to the comments are provided below.

#3-1. The authors claimed the effect of Ag-C interlayer as "The use of the Ag-C interlayer promoted the stable plating of Li metal within the TiN NT cavities and ensured continuous interfacial contact between the SE and TiN NT." (page 6, line 92-94), and the Li deposition in Ag-C layer and inside the TiN structure was analyzed using in-situ OM and TOF-SIMS. However, it is hard to elucidate the effect of Ag-C in this study. To clarify the effect of Ag-C for Li deposition, it is necessary to compare the performance of TiN NT without Ag-C. The improvement in cycling can be just ascribed to the effect of minimizing volume changes.

Response: We express our gratitude to the reviewer for their valuable comments. Garnet-type oxide solid electrolytes possess numerous advantages including high ionic conductivity, a wide electrochemical stability window, chemical stability, etc. making them a promising candidate for SSB electrolyte materials. However, their inherent characteristics, such as high rigidity and a rough surface, pose potential challenges when directly interfaced with anode materials of corresponding surface roughness. Such scenario may lead to the formation of Li dendrites at point contacts, impeding the effective transfer of Li ions. For the case of TiN NT, a requisite level of mechanical strength is necessary to withstand the internal stresses arising from Li metal deposition. Consequently, the direct contact between TiN NT and LLZTO introduces a surface disparity, potentially elevating the risks of contact loss and short circuits. To mitigate these concerns associated with surface irregularities, the introduction of an interlayer proves instrumental. The incorporation of an interlayer thus aims to minimize the likelihood of interfacial issues on the LLZTO/TiN NT interface, thereby fostering more robust and reliable electrochemical environment. Despite this importance, we agree that the effect of Ag-C interlayer was not clearly elucidated in the manuscript. Therefore, we analyzed the performance of TiN NT without Ag-C and conducted an electrochemical performance test to manifest the efficacy of the Ag-C interlayer in improving cycle stability. Further details and responses to specific points raised by the reviewer are provided in the follow-up comments.

#3-1-A. The full cell cycling of only TiN NT and Ag-C with TiN NT should be compared.

Response: We again thank the reviewer for their helpful comment. In response to the reviewer's suggestion, we constructed a TiN NT incorporated full-cell without

using the Ag-C interlayer and conducted an electrochemical test. However, as shown in the voltage profile result in Figure R14, the cell suffered a short-circuit during the charging process and did not operate thereafter. In previous report, it was demonstrated that the Ag-C interlayer facilitates the facile diffusion of reduced Li and directs Li metal deposition at the Ag-C/current collector interface (*Nat. Commun.*, 2023, 14, 782). Accordingly, in our system, the utilization of Ag-C has proven instrumental in achieving uniform interfacial contact at the LLZTO/TiN NT interface while also guiding the deposition of reduced Li into the TiN NT. This dual functionality of the Ag-C interlayer has contributed to the overall efficacy of the system in terms of both interfacial stability and enhanced Li deposition.

Figure R14. Charge/discharge voltage profile of the TiN NT incorporated full-cell without Ag-C interlayer. (This Figure is included as Supplementary Fig. 11 in the revised manuscript).

Page 9

Original manuscript

Recent studies have shown that a Ag-C interlayer serves as an effective dendrite protection layer for both sulfide and oxide SEs by directing Li deposition towards the Ag-C/current collector interface^{18,27}. However, compared to sulfide SE-based anode-free cells, the use of a Ag-C interlayer alone in oxide SE-based

anode-free (or Li-free) cells has not yet produced satisfactory cyclability^{17,32}. This can be attributed to the brittle and inelastic nature of oxide SE, which makes it susceptible to interfacial disintegration and mechanical failure from recurring local stress and volume changes during cycling³³⁻³⁵. Hence, accommodating reduced Li within the free volume space of the TiN NTs can effectively mitigate anisotropic strain caused by the formation of a Li deposition layer between interfaces. It is also important that other components in the cell undergo minimal volume change during charging/discharging to achieve stable operation of oxide SE-based AFSSBs. In this regard, operando analysis using optical microscopy (OM) was conducted to observe the Li propagation behaviour through the Ag-C interlayer during the charge/discharge process, while cross-section SEM was employed to examine the volume change of the Ag-C interlayer. Furthermore, the volume change suppression capability of the TiN NTs was examined by measuring the variations in the thickness of the full cell during cycling.

Revised manuscript

Recent studies have shown that a Ag-C interlayer serves as an effective dendrite protection layer for both sulfide and oxide SEs by directing Li deposition towards the Ag-C/current collector interface^{18,27}. However, compared to sulfide SE-based anode-free cells, the use of a Ag-C interlayer alone in oxide SE-based anode-free (or Li-free) cells has not yet produced satisfactory cyclability^{17,32}. This **limitation** can be attributed to the brittle and inelastic nature of oxide SE, which makes it susceptible to interfacial disintegration and mechanical failure from recurring local stress and volume changes during cycling³³⁻³⁵. **Nonetheless, the incorporation of Ag-C has proven instrumental in our system, as its role extends beyond guiding the deposition of reduced Li towards the negative electrode. It also plays a crucial role in achieving uniform interfacial contact at the LLZTO/TiN NT interface. This is particularly evident when TiN NT is used alone in the full-cell without Ag-C interlayer. As illustrated in the voltage profile result (Supplementary Fig. 11), the cell experienced an immediate short-circuit during the charging process and failed to operate**

thereafter. Thus, accommodating reduced Li within the free volume space of the TiN NTs necessitate the use of Ag-C interlayer to establish seamless interfacial contact at the LLZTO/TiN NT interface. To effectively mitigate the volume changes in oxide SE-based AFSSBs during cell operation, it is imperative to closely examine the alterations occurring at the interface and the bulk components of cell during charging/discharging process. In this regard, *operando* analysis using optical microscopy (OM) was conducted to observe the Li propagation behaviour through the Ag-C interlayer during the charge/discharge process, while cross-section SEM was employed to examine the volume change of the Ag-C interlayer. Furthermore, the volume change suppression capability of the TiN NTs was examined by measuring the variations in the thickness of the full cell during cycling.

#3-1-B. The Li deposition inside TiN NT for only TiN NT and Ag-C with TiN NT should be examined for comparison.

Response: We again express our gratitude to the reviewer for their valuable suggestion. As mentioned earlier, using TiN NT alone (without the Ag-C interlayer) leads to an immediate short circuit during the charging process, making it impossible to detect Li deposition inside TiN NT. In order to emphasize the role of Ag-C interlayer, we used the PEO+LiTFSi interlayer as a comparative example and examined their Li transport behavior via ex-situ SEM analysis. As illustrated in Figure R15a, b, when the PEO+LiTFSi interlayer is used, Li was observed to deposit not inside the TiN NT but rather precipitate above the TiN NT. In contrast, with TiN NT and the Ag-C interlayer, successful internal Li deposition within the nanotube was confirmed. The cross-sectional FIB SEM and EDS elemental mapping image of the charged cell (Figure R16) reveals distinct regions corresponding to TiN NT (Figure R16b, c) and Ag-C layers (Figure R16e, f), where the presence of O elements in both TiN NT and Ag-C regions indicates the existence of Li (in oxide form due to air exposure) (Figure R16d). This implies the presence of Li_{bcc} deposition within TiN NT and AgLi alloy within the Ag-C, respectively (Figure R16g). This result accentuates the negligible amount of residual lithium metal at the interface and highlights the effective Li_{bcc}

accommodating capability of TiN NT. Figure R16 and its related discussions are reflected in the revised manuscript (page 8) and Supplementary information (Supplementary Fig. 10).

Figure R15. a), b) Cross-sectional SEM images of the TiN NT and PEO+LiTFSi interlayer incorporated cell after charging. (This Figure is for reviewers only).

Figure R16. a) Cross-sectional FIB-SEM image, and b-g) Corresponding EDS

elemental mapping images of the TiN NT-incorporated AFSSB after charging. (This Figure is included as Supplementary Fig. 10 in the revised manuscript).

Page 8

Original manuscript

In contrast, for the case of 100% SOC, a strong Li^+ signal was detected throughout the sputtering time, indicating that Li metal was successfully deposited inside the TiN NT capillaries over the course of the charging process. These results suggest that the TiN NT anode can effectively accommodate Li metal inside its porous structure, mitigating volume expansion in the interlayer between the Ag-C and the current collector. This enables strain-free operation, as minimized volume change can prevent associated mechanical stress and deformation applied to the cell during the charging/discharging process.

Revised manuscript

In contrast, for the case of 100% SOC, a strong Li^+ signal was detected throughout the sputtering time, indicating that Li metal was successfully deposited inside the TiN NT capillaries over the course of the charging process. These results suggest that the TiN NT anode can effectively accommodate Li metal inside its porous structure, mitigating volume expansion in the interlayer between the Ag-C and the current collector. This enables strain-free operation, as minimized volume change can prevent associated mechanical stress and deformation applied to the cell during the charging/discharging process. Furthermore, the cross-sectional focused ion beam (FIB)-SEM analysis of the charged cell was performed to detect any residual lithium metal at the interface (Supplementary Fig. 10a). The EDS elemental mapping images reveal distinct regions corresponding to TiN NT (Supplementary Fig. 10b, c) and Ag-C layers (Supplementary Fig. 10e, f), characterized by a homogeneous distribution of their component elements. Evidently, the presence of O element in both FIB-milled TiN NT and Ag-C regions indicates the existence of Li (in oxide form due to air exposure), which are expected to manifest as Li_{bcc} deposition within

TiN NT and AgLi alloy within the Ag-C, respectively (Supplementary Fig. 10d, g). This result accentuates that the amount of residual lithium metal at the interface is negligible and also highlights the effective Li_{bcc} accommodating capability of TiN NT.

#3-2. In relation to question #1, the author also stated “Additionally, a Ag-C interlayer is embedded between the LLZTO and TiN NT architecture to facilitate uniform Li deposition across the nanotubes by channelling the redox reaction at the interface between the Ag-C interlayer and TiN NT27-29.” (page 4, line 67-70). Please clarify this statement. If this statement indicates that the redox reaction occurs at the interface between Ag-C and TiN or in Ag-C and TiN, this statement should be corrected. The redox reaction does not occur at the interface between Ag-C interlayer and TiN NT, but at the surface of LLZTO (or at the interface between Ag-C interlayer and LLZTO). The reduced Li metal move toward the current collector side by diffusional creep.

Response: We thank the reviewer for their elaborate and helpful comment. We acknowledge the need for clarification in the existing statement. As the Ag-C interlayer constitutes a carbon material functioning as a mixed electronic-ionic conductor, the redox reaction takes place at the interface between the solid electrolyte (LLZTO) and Ag-C interlayer (*Nat. Energy* 2020, 5, 299, *ACS Energy Lett.* 2023, 8, 9). In our previous report, we substantiated this phenomenon through a combination of DFT calculations and experimental evidence, confirming that Li, reduced at the LLZTO and Ag-C interlayer interface, diffuses toward the current collector via the Ag-C interlayer (*Nat. Commun.* 2023, 14, 782). In this study, the Ag-C interlayer in our LLZTO/Ag-C/TiN NT system also facilitates the rapid movement of the reduced Li toward the current collector during the charging process. This successful transport results in the deposition of Li inside the TiN NT rather than at the LLZTO/Ag-C interface. Therefore, we have modified the statement as follows:

Page 4

Original manuscript

Additionally, a Ag-C interlayer is embedded between the LLZTO and TiN NT architecture to facilitate uniform Li deposition across the nanotubes by channelling the redox reaction at

the interface between the Ag-C interlayer and TiN NT²⁷⁻²⁹.

Revised manuscript

Additionally, an Ag-C interlayer is embedded between the LLZTO and TiN NT architecture to facilitate the rapid transport of the reduced Li metal towards the TiN NT, thereby promoting uniform Li deposition within the TiN NT²⁷⁻²⁹.

#3-3. Using TiN NT imposes limitation on the achievable capacity. Currently, the diameter of the inner tubes in the structure is 100 nm. Using a larger diameter structure and/or porosity by thinning the tube wall could potentially increase lithium storage capacity.

Response: We thank for your helpful comment. Controlling the structure and porosity of the 3D MIEC architecture is crucial because it can maximize Li storage capacity and the energy density of the full cell. The tubular architecture is considered the most desirable 3D MIEC architecture as it can reduce the diffusion distance of Li atoms and stress accumulation. However, as the pore size of the nanotube increases during the anodization process, the wall thickness inevitably becomes thinner, compromising the mechanical robustness of the nanotube architecture (*Isr. J. Chem.* 2010, 50, 453, *Microporous Mesoporous Mat.* 2011, 147, 87-92). Notably, volume expansion occurs during lithiation/delithiation cycling, and due to the electrochemical aggressiveness and mechanical stress of Li metal, the nanotube wall must ensure sufficient strength (*Chem* 2020, 6, 2878-2892, *J. Electrochem. Soc.* 2019, 166, A89).

#3-3-A. Is it possible to control the diameter or wall thickness? Have you tried before?

Response: We formed a TiO₂ nanotube array through an anodization process of Ti foil, followed by the synthesis of TiN NTs through heat treatment under NH₃ conditions. In the anodization process, the length, pore size, and wall thickness of the nanotube array can be adjusted by varying the applied voltage, time, F⁻ ion concentration of the electrolyte, and temperature (*Energy Environ. Sci.* 2012, 5, 6506, *J. Phys. Chem. C* 2012, 116, 18669-18677, *Electrochem. Acta* 2007, 52, 8044-8047). We synthesized TiN NTs with a pore size ranging from 30 to 100 nm by controlling the applied voltage and time (Figure R17).

As the applied voltage and time increase, the pore size increases while the wall thickness decreases. If the anodization process continues beyond a certain point, the wall thickness becomes too thin to maintain the nanotube structure (Figure

R18). This is attributed to the continuous dissolution of the upper part of the tube due to F^- ions during prolonged anodization, leading to a weakening of mechanical strength (*Angew. Chem. Int. Ed.* 2011, 50, 2904, *Mater. Sci. Eng. R-Rep.* 2013, 74, 377-406). Therefore, a TiN NT structure is designed by carefully considering the trade-off between increased porosity and lithium storage capacity and the decrease in mechanical strength.

Figure R17. Top-view SEM images of the TiN NT prepared using various second-step anodization processes at a) 20 V for 1 h, b) 40 V for 1 h, and c) 50 V for 1 h. (This Figure is for reviewers only).

Figure R18. a) Top-view and b) Cross-sectional SEM images of the TiN NT prepared using a second-step anodization process at 70 V for 1 h. (This Figure is for reviewers only).

#3-3-B. Furthermore, within the current evaluation conditions, does the TiN NT layer expand in diameter? Is there any interfacial Li deposition at the interface between Ag-C and TiN or on the current collector? The cross-sectional images after Li deposition might help.

Response: Thanks for your valuable suggestion. To verify the impact of Li metal plating/stripping on the structure of the TiN NT wall, we conducted the cross-sectional SEM analysis of TiN NT, examining the diameter of the nanotubes

before and after the cycle (Figure R19). The results indicate that the diameter of TiN NT remains unchanged even after the cycling. This confirmed the robust mechanical strength of our TiN NT architecture, demonstrating its ability to withstand the stress associated with the Li plating/stripping process.

In addition, we verified the presence of deposited Li at the Ag-C and TiN NT interface through cross-sectional FIB-EDS analysis of the full-cell after charging. As shown in Figure R16, it was confirmed that no interfacial layer formed between the Ag-C interlayer and TiN NT after charging. Notably, the O 1s signal was uniformly observed not only in the Ag-C layer but also within the TiN NT, confirming successful Li penetration into the void of the TiN NT. This result indicates that the Li, which moved through the Ag-C layer, successfully deposited inside the TiN NT without precipitating at the interface. Furthermore, it is consistent with the SEM analysis results demonstrating the presence of Li metal inside the TiN NT after charging, as shown in Supplementary Fig. 8. Figure R19 and its related discussions are reflected in the revised manuscript (page 7) and Supplementary information (Supplementary Fig. 9).

Figure R19. Cross-sectional SEM images of TiN NT at a) before cycling, and b) after cycling. (This Figure is included as Supplementary Fig. 9 in the revised manuscript).

Page 7

Original manuscript

Consequently, the Li metal is fully deposited inside the tubules when the SOC

reaches 100% (Fig. 2b). The cross-sectional scanning electron microscopy (SEM) image of the fully charged TiN NT also confirmed that Li metal was successfully deposited inside the tubular structure (Supplementary Fig. 7). These results were consistent with those of previous studies showing that diffusional creep is a major lithiation mechanism in an MIEC tubular matrix with 100 nm wide and 10–100 μm deep channels in an SE-based battery system^{19,26}.

Revised manuscript

Consequently, the Li metal is fully deposited inside the tubules when the SOC reaches 100% (Fig. 2b). The cross-sectional scanning electron microscopy (SEM) image of the fully charged TiN NT also confirmed that Li metal was successfully deposited inside the tubular structure (Supplementary Fig. 8). To assess the impact of Li metal plating/stripping on the TiN NT wall structure, we conducted the cross-sectional SEM analysis of TiN NT, examining the diameter of the nanotubes before and after the cycling (Supplementary Fig. 9). The results indicate that the diameter of TiN NT remains unchanged even after the cycling process. This confirmed the robust mechanical strength of our TiN NT architecture, demonstrating its ability to withstand the stress associated with the Li plating/stripping process. These findings align with prior studies, showing that diffusional creep serves as a major lithiation mechanism in a MIEC tubular matrix featuring 100 nm wide and 10–100 μm deep channels in an SE-based battery system^{19,26}.

#3-4. In ToF-SIMS data, the amount of Li within TiN NT progressively decrease toward the current collector side. This indicate that the TiN NT is not fully filled with Li. What is the maximum capacity when TiN NT is completely filled? Is there any interfacial deposition between Ag-C and TiN NT layer when you increase the capacity before TiN NT is completely filled? Then, the diffusional creep through TiN should be enhanced for fully accommodating Li inside TiN NT. This point with the related data should be addressed in the manuscript.

Response: We thank you for your insightful comment. We calculated the maximum areal capacity of TiN NT to be 2.473 mAh/cm² based on the porosity of TiN NT. Using this maximum

areal capacity, we designed a full cell configuration with a TiN NT anode ($\text{\O} 5\text{mm}$) and NCM 333 cathode ($\text{\O} 4\text{mm}$). Since the maximum capacity of the anode (0.485 mAh) exceeds the designed cathode capacity (0.402 mAh), there is a concern suggesting that the inside of the TiN NT may not be completely filled with Li metal even after fully charging. In particular, designing the cathode capacity to exceed the maximum capacity of the anode could result in excess Li remaining after filling the TiN NT, leading to plating between Ag-C and TiN NT and subsequent volume expansion. Additionally, as Li metal is deposited inside the tubular structure, the internal pressure of the tubular structure locally increases, significantly reducing the mechanical stability of the TiN NT structure (*Nature*, 2020, 578, 251, *Nat. Commun.*, 2016, 7, 10146). Therefore, the designed cathode capacity must be smaller than the maximum capacity of the anode.

Nevertheless, during the charging process, no Li interfacial deposition was observed at the interface between the Ag-C interlayer and TiN NT (**Figure R16**), indicating that Li was deposited inside the TiN NT rather than at the Ag-C/TiN NT interface. Relevant discussions are reflected on **page 6** in the revised manuscript.

Page 6

Original manuscript

The areal capacity of the TiN NT anode was estimated based on the porosity of the TiN NT architecture (Supplementary Fig. 5), and it was designed to exceed the NCM333 areal capacity (3.2 mAh cm^{-2}) to prevent Li overflowing and plating onto the TiN NT surface.

Revised manuscript

The maximum areal capacity of the TiN NT anode based on the porosity of the TiN NT and the specific capacity of Li metal was calculated as 2.473 mAh/cm^2 (Supplementary Fig. 5). To prevent Li overflow and plating onto the TiN NT surface, we designed the capacity of the NCM 333 cathode not to exceed the maximum capacity of the TiN NT anode.

#3-5. The Pre_Li_Ag-C is formed by simple compression using CIP with halo Li foil, as mentioned in page 9, line 155-156. I recommend adding information about this prelithiation process (such as amount of reacted lithium foil and reaction time) to the methods section.

Response: We appreciate your kind suggestion. The halo Li ring (exterior \O : 12 mm, interior

Ø: 8 mm, thickness: 20 µm) was attached to the Ag-C surface using 250 MPa CIP for 3 min during the pre-lithiation process. Subsequently, it is maintained at 45°C for 7~12 hours to ensure sufficient pre-lithiation of the Ag-C interlayer before use. The details of the pre-lithiation process have been included in the method section in the revised manuscript.

Page 22

Revised manuscript

Cell assembly and electrochemical characterizations.

We employed a quasi-all-solid-state cell. In each quasi-all-solid-state cell, an ionic liquid, and a solid oxide electrolyte (LLZTO) were used as the cathode and anode electrolytes, respectively. A Ag-C layer coated on a 10-µm-thick stainless steel (SUS) foil was prepared using a mixture of carbon black powder (99.7%, average particle size = 38 nm, Asahi carbon) and Ag nanoparticles (D50 = 60 nm).²⁷ Ag and carbon black powder were mixed in a weight ratio of 1:3 in N-methylpyrrolidone (Sigma-Aldrich) with 7 wt% polyvinylidene fluoride (Solvay) as a binder under constant stirring (1000 rpm) for 30 min using a mixer (Thinky Corporation, AR-100). The resulting slurry was then coated on SUS foil using a screen printer and dried in air at 80 °C for 20 min. The coated Ag-C layer was further dried under a vacuum at 100 °C for 12 h. The Ag-C layer was attached as an anode interlayer on the acid-treated LLZTO surface via cold-isostatic pressing (CIP) under 250 MPa. After the SUS foil was peeled, a TiN NT or Cu foil was attached to the Ag-C surface via 250 MPa CIP.

Commercially available $(\text{Li}_{1+x}(\text{Ni}_{0.33}\text{Co}_{0.33}\text{Mn}_{0.33}))_{1-x}\text{O}_2$ (NCM333, loading capacity: 3.2 g/cc, active material: 96 wt%; thickness: 50 µm; Samsung SDI) was employed as a cathode and as a current collector, Al foil (9 µm foil, Nippon Foil Mfg. Co., LTD) was used as received. A N-methyl-N-propyl pyrrolidinium bis(fluorosulfonyl) imide (Pyr13FSI, 99.9%, water content <20 ppm, Kanto Chemical Co. Inc.) ionic liquid mixed with a 2 M lithium

bis(fluorosulfonyl)imide (LiFSI, 99.9%, water content <10 ppm) salt was used as the catholyte. The catholyte solution (20 wt% relative to the cathode) was infiltrated into the cathode in a dry room (dew point, $-60\text{ }^{\circ}\text{C}$), followed by maintaining a vacuum state for 2 h. When the residual solution on the cathode surface was removed using Kimwipes, the solution uptake by the cathode was $\sim 7\text{ wt}\%$. We placed the ionic-liquid-infiltrated cathode on the cathode side of the LLZTO. For Ag-C prelithiation, a halo Li was prepared by punching the Li-Cu foil (with a Li thickness of $20\text{ }\mu\text{m}$) into a ring with exterior and interior diameters of 12 and 8 mm, respectively. Subsequently, the prepared halo Li was affixed to the Ag-C coated side of the LLZTO via 250 MPa CIP for 3 min, followed by incubation in an oven set at $45\text{ }^{\circ}\text{C}$ for 7~12 h. The entire cell components were then assembled in a CR2032 coin cell, including a spring and two 0.5T disks (as shown in Supplementary Fig. 15), resulting in an internal pressure of approximately 0.6 MPa after clamping.

#3-5-A. Does the halo Li foil used in this prelithiation reaction participate entirely in the reaction, leaving no residual lithium layer?

Response: Thank you for your insightful question. Due to the strong adhesion of halo Li foil with the Ag-C interlayer, accurately measuring the amount of Li used in pre-lithiation becomes challenging, as it cannot be easily separated from the Ag-C interlayer. To address this issue, we employed titration gas chromatography (TGC) analysis to indirectly assess the remaining halo Li after the pre-lithiation process (Figure R20 and R21). Calculating the residual Li metal by measuring H_2 generated upon reaction with H_2O for each sample confirmed that approximately 0.405 mg of excess Li metal remained in the halo Li of LLZTO/Pre_Li_Ag-C sample after pre-lithiation. This result indicates that only a specific portion of halo Li actively participates in the pre-lithiation of the Ag-C interlayer.

Despite this, subsequent measurements on the LLZTO/Pre_Li_Ag-C sample after the charging/discharging cycle revealed that the amount of Li metal remained consistent with the pre-cycle sample. This observation suggests that excess halo

Li does not impact the charging/discharging process of the full cell.

Figure R20. Calibration curve of the detected H₂ area as a function of Li metal amount. (This Figure is for reviewers only).

Figure R21. Generated H₂ peaks after reactions with H₂O for (a) LLZTO/Ag-C, (b) pristine halo Li, (c) LLZTO/Pre_Li_Ag-C (before cycle), and (d) LLZTO/Pre_Li_Ag-C (after cycle). (This Figure is for reviewers only).

#3-5-B. In Fig. 3d, the cross-sectional SEM image of “As-is” shows a significant amount of lithium layers with a thickness of over 10 μm before cycling. What is the role of this lithium layer at this stage? Why does it exist? If extra Li layer exists during cycling, it should be not mentioned as “pre-lithiated”. It is not a Li-free or anode-free system anymore.

Response: Thank you for your question. Firstly, I would like to elucidate the cell structure shown in Fig. 3d. This specific cell was employed in the experiment to confirm both the charging capacity contribution and independent volume change capability of the Ag-C interlayer. It is important to clarify that Li metal was used as the anode to investigate the role of the Ag-C interlayer. Consequently, the Li layer shown in Fig. 3d is irrelevant to the full cell configuration employing TiN NT as the anode. Notably, in our full-cell configuration, there is no additional Li layer present between TiN NTs and the Ag-C interlayer (Fig. 1c).

#3-5-C. What are the differences between Ag-C/Li and Pre_Li_Ag-C as compared in Fig. 4?

Response: Thank you for your question. In Fig. 4, Ag-C/Li represents a cell with Li metal as the anode, used to assess volume changes during the cycling of the Li metal anode and the anode-free (Li-free) configuration (Cu or TiN NT as the anode). Pre_Li_Ag-C refers to the Ag-C interlayer after the pre-lithiation process, and the halo Li used in pre-lithiation process does not affect cycling (Response #3-5-A).

#3-6. In Fig. 5e, the authors compared Cu and TiN NT. How do you drive the numerical values of the impedance (the impedance difference mentioned in page 14, line 243-245)? I recommend providing an equivalent circuit for the EIS data.

Response: We thank the reviewer for their valuable suggestions. Following the suggestions of the reviewer, we calculated the ohmic resistance (R_b), interfacial resistance (R_{ct}), and Warburg resistance (W_R) by complex nonlinear least-squares fitting of the experimental impedance spectra in Fig. 5e and Supplementary Fig. 13 based on a given equivalent circuit. We have added images of the equivalent circuit and listed the resultant fitting values to Supplementary Table 2 and 3 to facilitate a better comparison of R_b , R_{ct} , and W_R .

Table R3. Fitting parameters determined from CNLS fitting of the measured impedance spectra in Supplementary Fig. 18. (This Table is included as Supplementary Table 2 in the revised manuscript).

[Symmetric cell (Pristine)]

Fitting parameters	Anode			
	Cu		TiN NT	
	Fit values	Error (%)	Fit values	Error (%)
R_b [Ω cm ²]	18.67	0.36	18.85	0.42
R_1 [Ω cm ²]	5.76	2.52	3.96	2.81
CPE_{1-T}	6.12×10^{-6}	19.17	3.17×10^{-5}	20.44
CPE_{1-n}	0.86	2.38	0.77	3.00
chi-square	5.1×10^{-4}		9.68×10^{-4}	

[Symmetric cell (After cycle)]

Fitting parameters	Anode			
	Cu		TiN NT	
	Fit values	Error (%)	Fit values	Error (%)
R_b [Ω cm ²]	18.08	0.57	18.44	0.48
R_1 [Ω cm ²]	9.03	1.65	5.56	2.27
CPE_{1-T}	1.36×10^{-5}	14.26	2.92×10^{-5}	17.94
CPE_{1-n}	0.75	2.02	0.74	2.70
chi-square	1.78×10^{-3}		1.45×10^{-3}	

Table R4. Fitting parameters determined from CNLS fitting of the measured impedance spectra in Fig. 5e. (This Table is included as Supplementary Table 3 in the revised manuscript).

[60 °C]

Fitting parameters	Anode			
	Cu		TiN NT	
	Fit values	Error (%)	Fit values	Error (%)
R_b [Ω cm ²]	7.34	0.11	5.0	0.15
R_l [Ω cm ²]	14.10	0.93	12.75	0.93
R_2 [Ω cm ²]	2.54	2.76	2.41	2.35
$R_{ct} = R_l + R_2$ [Ω cm ²]	16.64		15.16	
CPE ₁ -T	1.85×10^{-5}	2.54	1.10×10^{-5}	2.67
CPE ₁ -n	0.76	0.33	0.87	0.31
W _R [Ω cm ²]	14.44	10	63.87	11.22
W-T	0.076	10.19	0.25	13.82
W-n	0.40	0.79	0.37	1.38
CPE ₂ -T	4.8×10^{-6}	15.85	2.44×10^{-6}	16.07
CPE ₂ -n	0.89	4.05	0.95	3.44
chi-square	1.35×10^{-4}		2.19×10^{-4}	

[25 °C]

Fitting parameters	Anode			
	Cu		TiN NT	
	Fit values	Error (%)	Fit values	Error (%)
R_b [$\Omega \text{ cm}^2$]	31.06	0.16	22.01	0.2
R_1 [$\Omega \text{ cm}^2$]	25.98	4.39	12.34	3.98
R_2 [$\Omega \text{ cm}^2$]	81.39	1.21	49.74	2.79
$R_{ct} = R_1 + R_2$ [$\Omega \text{ cm}^2$]	107.37		62.08	
CPE ₁ -T	9.3×10^{-6}	8.95	3.17×10^{-6}	12.6
CPE ₁ -n	0.73	1.11	0.87	1.48
W_R [$\Omega \text{ cm}^2$]	61.97	3.85	63.87	3.95
W-T	0.29	5.56	0.25	6.3
W-n	0.40	0.99	0.37	1.92
CPE ₂ -T	5.75×10^{-6}	2.37	2.44×10^{-6}	5.95
CPE ₂ -n	0.86	0.07	0.95	1.18
chi-square	3.38×10^{-4}		2.60×10^{-4}	

Figure R22. Nyquist plots of AC-impedance spectra obtained from the symmetric cells with different anodes (Cu vs. TiN NT) before and after cycle (inset: equivalent circuit). a) Before cycle and b) After cycle. (This Figure is included as Supplementary Fig. 18 in the revised manuscript).

Figure R23. Nyquist plots of AC-impedance spectra obtained from the AFSSB full-cells with different anodes (Cu vs. TiN NT) at different temperature conditions (inset: equivalent circuit). a) 25 °C and b) 60 °C. (This Figure is included as Fig. 5e and Supplementary Fig. 20 in the revised manuscript).

Page 15

Original manuscript

The electrochemical impedance spectroscopy (EIS) result (Supplementary Fig. 13) also revealed that the total resistance could be reduced from $3.3 \Omega \text{ cm}^2$ to $2.6 \Omega \text{ cm}^2$ for the pristine cell and from $8.4 \Omega \text{ cm}^2$ to $5.0 \Omega \text{ cm}^2$ for the cycled cell by employing Pre_Li_Ag-C/TiN NT.

Revised manuscript

Moreover, the electrochemical impedance spectroscopy (EIS) measurement of the same symmetric cells before and after cycle was conducted to evaluate their change in the cell impedance (Supplementary Fig. 18a, b). The Nyquist plot and corresponding fitting result (Supplementary Table 2) revealed that the TiN NT symmetric cell exhibits much lower interfacial resistance compared to its Cu counterpart before cycle (3.96 vs. $5.76 \Omega \text{ cm}^2$) and the trend persists after the cycle (5.56 vs. $9.03 \Omega \text{ cm}^2$), signifying the efficacy of the TiN NT in maintaining the interfacial contact during cell operation.

Original manuscript

~ Additionally, we conducted EIS to analyze the effect of the TiN NTs on the impedance of the cell and compared it with that of the Cu current collector under two temperature conditions (60 °C and 25 °C). Notably, the impedance difference between the TiN NT and Cu installed full cells was negligible at 60 °C (9.2 Ω cm² vs. 11.2 Ω cm²), while there was a significant difference in the impedance at 25 °C (55.3 Ω cm² vs. 108.7 Ω cm²) (Fig. 5e). This large disparity can be ascribed to the increase in interfacial resistance due to the reduced ionic conductivity of LLZTO SE at room temperature. Thus, this signifies that the current collector interface is a dominating factor for cell operation at low temperature (25 °C).

Revised manuscript

~ Additionally, we conducted EIS analysis on TiN NT and Cu installed full-cells at both 25 and 60 °C to comparatively examine their impact on cell impedance under difference temperature conditions. Fig. 5e and Supplementary Fig. 20 shows the Nyquist plots of the impedance spectra consisting of two depressed arcs, which overlap in the high frequency range, Warburg impedance, and a capacitive line (blocking region) in the low-frequency range. The equivalent circuit was implemented to quantitatively analyze the experimental impedance spectra, and the complex nonlinear least-squares (CNLS) fitting method was used to determine the bulk ionic resistance (R_b) and interfacial resistance ($R_{ct} = R_1 + R_2$) (the values are listed in Supplementary Table 3). Notably, a more pronounced difference in R_b and R_{ct} values between the TiN NT and Cu full-cells occurs at 25 °C (than at 60 °C) due to the overall increase in bulk and interfacial resistance associated with the lower ionic conductivity of the LLZTO SE at room temperature. Amidst of this limited ionic conduction, the impedance contribution at the Ag-C and the current collector interface becomes dominant. This is corroborated by the significantly lower R_b and R_{ct} values (22.0 and 62.1 Ω cm²) of the TiN NT full-cell compared to its Cu counterpart (31.1 and 107.4 Ω cm²), which can be ascribed to superior charge-transfer kinetics and interfacial stability at the Ag-C|TiN NT interface.

#3-7. The full cell system is not actually an all solid-state battery because the cathode was impregnated with ionic liquid. To clearly indicate this, the “solid-state batteries” in the title should be revised to “hybrid solid-state battery”.

Response: We appreciate your helpful comment. Although we mentioned the use of an ionic liquid in the cathode in the Experimental section (page 21, line15) and referred to it as a quasi-all-solid-state cell, we agreed that the 'solid-state batteries' in the title might still confuse readers. Therefore, we have modified the title as follows:

Page 1

Original title

Strain-free anode architecture enabled by interfacial diffusion creep for anode-free solid-state batteries with garnet-type solid electrolyte

Revised title

Strain-free anode architecture enabled by interfacial diffusion creep for anode-free quasi-all-solid-state batteries with garnet-type solid electrolyte

REVIEWERS' COMMENTS

Reviewer #1 (Remarks to the Author):

The reviewer appreciates the efforts authors have placed on addressing all questions, which has resulted in a higher quality manuscript. It is suggested to utilize the Distribution of Relaxation Time (DRT) tools to more effectively separate the electrochemical processes during impedance analysis. Furthermore, the term "quasi-all-solid-state batteries" in the revised manuscript is not very clear. It is also recommended to include a scale bar in the supplementary movie for a more precise demonstration.

Reviewer #3 (Remarks to the Author):

The authors addressed the concerns adequately, and revised the manuscript accordingly. It may be suitable to be accepted as it is.

Point-by-point responses to Reviewers' comments

Manuscript ID: NCOMMS-23-38332A

Reponse Letter Contents

Response to Reviewer #1.....Page 2
Response to Reviewer #3.....Page 3

Response to Reviewer #1

[Remarks] The reviewer appreciates the efforts authors have placed on addressing all questions, which has resulted in a higher quality manuscript. It is suggested to utilize the Distribution of Relaxation Time (DRT) tools to more effectively separate the electrochemical processes during impedance analysis. Furthermore, the term "quasi-all-solid-state batteries" in the revised manuscript is not very clear. It is also recommended to include a scale bar in the supplementary movie for a more precise demonstration.

Response: We sincerely appreciate the reviewer for your specific and valuable comments aimed at improving our manuscript. Firstly, we attempted to utilize DRT tools to separate the electrochemical process during impedance analysis more effectively. However, applying DRT tools to solid-state batteries proved challenging, as it required extensive detailed analysis and interpretation to separate their contribution to impedance in the full cell (*Joule* 2022, 6, 1172-1198; *ChemElectroChem* 2021, 8, 1930-1974; *J. Power Sources* 2023, 556, 232450). There are concerns that this may deviate from the scope of our paper. Nevertheless, we still believe that impedance analysis using the DRT tool is important for understanding the electrochemical process of solid-state batteries, and we will address this through follow-up research. Thanks again to the reviewer for your valuable suggestion.

In addition, we renamed the term "quasi-all-solid-state batteries" to "quasi-solid-state batteries" for clarity, referring to existing papers utilizing ionic liquid catholytes (*Adv. Energy Mater.* 2017, 7, 1601196; *Energy Storage Mater.* 2023, 63, 103062). Lastly, we included a scale bar in the revised Supplementary Movie 1. The related discussions are reflected in the revised manuscript and Supplementary Movie.

Response to Reviewer #3

[Remarks] The authors addressed the concerns adequately, and revised the manuscript accordingly. It may be suitable to be accepted as it is.

Response: We sincerely appreciate the reviewer for constructive and positive comments on our revised manuscript.